**A Synthesis of Global Streamflow characteristics, Hydrometeorology, and**
**catchment Attributes (GSHA) for Large Sample River-Centric Studies**
Ziyun Yin[1], Peirong Lin[1,2*], Ryan Riggs[3], George H. Allen[4], Xiangyong Lei[1], Ziyan
Zheng[5,6], Siyu Cai[7]
1. Institute of Remote Sensing and GIS, School of Earth and Space Sciences, Peking University
2. International Research Centre for Big Data for Sustainable Development Goals, Beijing, China
3. Department of Geography, Texas A&M University, Texas, USA
4. Department of Geosciences, Virginia Polytechnic Institute and State University, Virginia, USA
5. Key Laboratory of Regional Climate-Environment Research for Temperate East Asia, Institute of
Atmospheric Physics, Chinese Academy of Sciences, Beijing, China
6. University of Chinese Academy of Sciences, Beijing, China
7. State Key Laboratory of Simulation and Regulation of Water Cycle in River Basin, China Institute
of Water Resources and Hydropower Research, Beijing, China
* *Correspondence to*: Peirong Lin (peironglinlin@pku.edu.cn)
Revised manuscript submitted to *ESSD, November 15th, 2023*
# Abstract
Our understanding and predictive capability of streamflow processes largely rely on high-
quality datasets that depict a river's upstream basin characteristics. Recent proliferation of large
sample hydrology (LSH) datasets has promoted model parameter estimation and data-driven
analyses of the hydrological processes worldwide, yet existing LSH is still insufficient in terms of
sample coverage, uncertainty estimates, and dynamic descriptions of anthropogenic activities. To
bridge the gap, we contribute the Synthesis of Global Streamflow characteristics, Hydrometeorology,
and catchment Attributes (GSHA) to complement existing LSH datasets, which covers 21,568
watersheds from 13 agencies for as long as 43 years based on discharge observations scraped from
web. In addition to annual and monthly streamflow indices, each basin's daily meteorological
variables (i.e., precipitation, 2 m air temperature, longwave/shortwave radiation, wind speed, actual
and potential evapotranspiration), daily-weekly water storage terms (i.e., snow water equivalence,
soil moisture, groundwater percentage), and yearly dynamic descriptors of the land surface
characteristics (i.e., urban/cropland/forest fractions, leaf area index, reservoir storage and degree of
regulation) are also provided by combining openly available remote sensing and reanalysis datasets.
The uncertainties of all meteorological variables are estimated with independent data sources. Our
analyses reveal the following insights: (i) the meteorological data uncertainties vary across variables
and geographical regions, and the revealed pattern should be accounted for by LSH users, (ii) ~6%
watersheds shifted between human managed and natural states during 2001-2015, e.g., basins with
environmental recovery projects in Northeast China, which may be useful for hydrologic analysis
that takes the changing land surface characteristics into account, and (iii) GSHA watersheds showed
a more widespread declining trend in runoff coefficient than an increasing trend, pointing towards
critical water availability issues. Overall, GSHA is expected to serve hydrological model parameter
estimation and data-driven analyses as it continues to improve. GSHA v1.1 can be accessed at
*https://doi.org/10.5281/zenodo.8090704* and *https://doi.org/10.5281/zenodo.10127757*. (Yin et al.,
42   2023).

# 1 Introduction

Climate change has posed profound challenges to the management of freshwater resources,
specifically riverine floods or water shortages (AghaKouchak et al., 2020; Thackeray et al., 2022).
The urgent need for flood and drought forecasting, water resources planning and management, all
call for high-quality streamflow predictions for basins worldwide to analyse global terrestrial water
conditions in a systematic view (Burges, 1998). The scarcity of hydrological observations has
brought challenges to these predictions (Belvederesi et al., 2022; Hrachowitz et al., 2013), thus the
development of computer models that allow for "modelling everything everywhere" (Beven &
Alcock, 2012) constitutes the backbone of hydrological studies. Existing studies have used
physically-based and data-driven models for streamflow simulation (Lin et al., 2018; Nandi &
Reddy, 2022; Zhang et al., 2020), with efforts to improve accuracy of prediction by combining both
(Cho & Kim, 2022; Razavi & Coulibaly, 2013). Yet the prediction of the magnitude, timing, and
trend of critical streamflow characteristics are still subject to multiple sources of errors and
uncertainties (Bourdin et al., 2012; Brunner et al., 2021).
Streamflow (Q) can be represented by the simple water balance equation involving
precipitation (P), evapotranspiration (ET), and water storage terms (S) denoted as $Q = P - ET - \Delta S$,
yet influencing factors of these components could bring uncertainties that cascade downstream.
Starting from the model assumptions to the data used to represent climate, soil water, ice cover,
topography and land use, as well as the less well-known processes such as human perturbations and
sub-surface flows (Benke et al., 2008; Wilby & Dessai, 2010), these complications impede our
understanding of streamflow processes across scales, which also limits the modelling and predictive
capability for streamflow. Thus, reducing the predictive uncertainties requires high-quality data with
massive samples capable of depicting each of the water balance components, as well as the natural
and anthropogenic factors involved (Gupta et al., 2014).
Efforts have been made to address the need for such kind of high-quality datasets on watershed-
scale hydro-climate and environmental conditions during the past couple of decades. One of the
earliest was the most widely used dataset generated for the Model Parameter Estimation Experiment
(MOPEX) project aimed at better hydrological modelling (Duan et al., 2006). Historical hydro-
meteorological data and land surface characteristics for over 400 hydrologic basins in the United
States were provided, which was fundamental to the progress in large sample hydrology (LSH)
(Addor et al., 2020; Schaake et al., 2006). Later the dataset was expanded to 671 catchments in the
contiguous United States (CONUS) and benchmarked by model results (Newman et al., 2015).
Based on these studies, the Catchment Attributes and Meteorology for Large-sample Studies
(CAMELS) dataset was developed, providing comprehensive and updated data on topography,
climate, streamflow, land cover, soil, and geology attributes for each catchment (Addor et al., 2017).
The CONUS CAMELS dataset soon became influential in LSH and has since inspired researchers
from Australia (Fowler et al., 2021), Europe (Coxon et al., 2020; Delaigue et al., 2022; Klingler et

al., 2021), South America (Alvarez-Garreton et al., 2018; Chagas et al., 2020), and China (Hao et al., 2021) to contribute their regional CAMELS. Another comprehensive regional LSH dataset for North America named the Hydrometeorological Sandbox - École de Technologies Supérieure (HYSETS) dataset, was also developed with larger sample size (14425 watersheds) and richer data sources compared with the CAMELS (Arsenault et al., 2020).

While these datasets are reliable data sources for regional studies, attempts on building global datasets have become the new norm in the era of big data to boost our analytical and modelling capability for the terrestrial hydrological processes. The HydroATLAS dataset integrated indices of hydrology, physiography, climate, land cover, soil, geology, and anthropogenic activity attributes for 8.5 million global river reaches (Lehner et al., 2022; Linke et al., 2019). A recent work combined a series of CAMELS datasets with HydroATLAS attributes into a new global community dataset on the cloud named Caravan, with dynamic hydro-climate variables and comprehensive static catchment attributes extracted on 6830 watersheds (Kratzert et al., 2023), which represents by far the most comprehensive synthesis of existing CAMELS. Another global-scale effort, the Global Streamflow Indices and Metadata archive (GSIM), incorporated dynamic streamflow indices and attribute metadata for topography, climate type, land cover, etc., for over 35000 gauges (Do et al., 2018; Gudmundsson et al., 2018), and the streamflow indices were updated to allow for trend analysis (Chen et al., 2023). A recent study filled in the discontinuity and latency of gauge records, and provided streamflow for over 45,000 gauges with improved data quality (Riggs et al., 2023). These global-scale datasets have been widely used in data-driven machine learning models (Kratzert et al., 2019a, 2019b; Ren et al., 2020), physical hydrological models (Aerts et al., 2022; Clark et al., 2021), and parameter estimation and regionalization studies (Addor et al., 2018; Fang et al., 2022).

Although the flourishment of LSH datasets has promoted comparative hydrological studies (Kovács, 1984) and large-scale hydrological modeling and analysis efforts, several challenges are still standing in the way of realizing the full potential of LSH. As briefly outlined in a recent review by Addor et al. (2020), current LSH datasets lack common standards, metadata and uncertainty estimates, and are insufficient in characterising human interventions. More specifically, the following major critical aspects still need attention from the LSH developers, which we attempt to address with GSHA (Yin et al., 2023). First, the majority of current datasets (especially those at a global scale) incorporated only one data source for each variable, while earth observations, reanalysis, satellite-based estimates are subject to uncertainties (Merchant et al., 2017; Ukhurebor et al., 2020). These uncertainties were rarely represented and may bring difficulties to the regionalization of model parameters (Beck et al., 2016), while also resulting in inconsistent conclusions. Second, anthropogenic activities including land use and land cover (LULC) changes, dam and reservoir building, etc., are critical drivers of shifts in streamflow statistical moments (Niraula et al., 2015). However, historical time series of watershed human modifications were rarely included in LSH datasets, which is particularly problematic for regions with rapid economic growth. Finally, although the most recent Caravan provided hydroclimate data for global watersheds, the samples are limited to the existing regional CAMELS which Caravan synthesizes. Therefore, plenty of room is left to increase data sample size and spatial coverage by revisiting the streamflow data acquisition process in a more comprehensive way.

To complement existing LSH datasets, we contribute the first version of a synthesis of Global Streamflow characteristics, Hydrometeorology, and catchment Attributes (GSHA v_1.0) for large-sample river-centric studies. GSHA features the following characteristics:

● Updated physical and anthropogenic descriptors of global rivers, covering streamflow
characteristics, hydrometeorological variables, and land use land cover changes for 21568
watersheds derived from gauged streamflow records from 13 agencies.
● Streamflow indices for data scarce regions, including those derived from 263 gauges in
China, are included.
● Extended temporal coverage for as long as 43 years (1979-2021), which varies regionally.
● Uncertainty estimates for the meteorological variables.
● Dynamic descriptors for the urban, forest, and cropland fractions, as well as reservoir
storage capacity to improve the representation of human activities in the basin.
With the above features, we expect GSHA to support hydrological model parameter estimation
and data-driven analysis of global streamflow as one of the most comprehensive LSH datasets
regarding sample size, variable dynamics, and uncertainty estimates. **Table 1** summarizes the
differences between GSHA and other prominent LSH datasets. Our paper is organized as follows.
Section 2 expands on **Table 1** and provides more details of the data included for GSHA. Section 3
introduces the data sources and methodologies involved in creating GSHA. Section 4 highlights the
key features of GSHA by conducting some analyses, followed by conclusions reached in Section 5.

**Table 1 Comparison of GSHA with other LSH datasets.** Note that we only include the CONUS
CAMELS dataset to represent regional LSH datasets for this comparison, as other regional CAMELS
share large similarity with CONUS CAMELS.

| Factors | CAMELS (eg. US) | HydroATLAS | Caravan | GSIM | GSHA |
|---|---|---|---|---|---|
| Spatial extent | Regional | Global | Global | Global | Global |
| Sample size | 671 | 8.5 million | 6830 | 35002 | 21568 |
| Time span | 1980–2015 | Static | 1981–2020 | 1806-2016 | 1979-2021 |
| Streamflow dynamics | Yes | No | Yes | Yes (statistical indices) | Yes (monthly and yearly statistical indices) |
| Meteorological time series | Yes | No | Yes | No | Yes |
| Multi data sources for meteorological variables | Yes | No | No | No | Yes (**with uncertainty estimates**) |
| Water storage dynamics | No | No | Only soil water dynamics | No | **Yes** |
| Land cover dynamics | No | No | No | No | **Yes** |
| Reservoir dynamics | No | No | No | No | **Yes** |
| Static attributes | Yes | Yes | Yes (from HydroATLAS) | Yes | Yes (from HydroATLAS) |

## 2 Dataset content of GSHA v1

In this section, the data fields, variables, and attributes included in GSHA are described in more details and summarized in **Table 2**. For the instructions of the data format, we provide a user manual along with the dataset (see readme.docx). GSHA includes yearly and monthly streamflow characteristics derived from daily discharge observations, meteorological variables (including precipitation, 2-m air temperature, long- and shortwave radiation, wind speed, actual and potential evapotranspiration (AET and PET)), daily or weekly water storage terms (4 layers of soil moisture, groundwater, and snow depth water equivalence), daily vegetation index (leaf area index (LAI)), yearly LULC characteristics (urban, cropland, and forest fraction), and yearly reservoir information (degree of regulation (DOR) and reservoir capacity). For each meteorological variable, multiple independent data sources are incorporated to provide uncertainty estimates. Static attributes like land physiography, soils, and geology are not additionally extracted, as similar efforts have been made by other researchers, so we directly matched our gauge locations to the HydroATLAS dataset (Lehner et al., 2022; Linke et al., 2019) by providing the river ID match table. Users can link the two to obtain these attributes.

**Watershed polygons:** GSHA includes 21568 watershed polygons delineated from the global gauges, which are stored as Esri Shapefile format. The ID and agency of each watershed is the same as the corresponding gauge ID, and the gauge latitude/longitude are in decimal degree. The area denotes the upstream drainage area of the gauge. Some of the IDs contain characters (such as '.', '-', etc.) inconsistent with the majority of IDs. For the convenience of the users, we unified these as underscores and stored the new file names as 'filename'. We also provide independent files summarizing basic information of the watersheds, including matched MERIT river reach COMID, upstream area, order and downstream river reach COMID, as well as verification with officially reported areas of the agencies.

**Streamflow indices:** GSHA publishes annual and monthly streamflow indices derived from daily streamflow data, including different percentiles, and mean/median/minimum/maximum. The frequency and durations of extremely high and low streamflow events are also provided. We also include numbers of zero observations and valid samples to allow flexible data screening by the users. The indices are stored as comma-separated values (CSV) files, with each watershed corresponding to one file. A complementary R package can be used to automatically download many of the gauge datasets is available at https://github.com/Ryan-Riggs/RivRetrieve (Riggs et al., 2023).

**Meteorological variables:** The meteorological variables selected are the most influential drivers for streamflow, which include precipitation, 2-m temperature, ET, radiation and wind speed. In main-stream land surface models, ET is a diagnostic variable derived from meteorological inputs and is not considered as meteorological forcing. However, as many hydrological models also use potential ET as an input variable, and model calibration sometimes involves actual ET (Immerzeel & Droogers, 2008), we include the two variables and place them into the meteorological variable category. For each variable, more than one data sources are used to allow for uncertainty analysis, which is provided on a yearly basis in an independent file.

**Natural water storage terms and land use/land cover change:** These include soil moisture,
snow water equivalent, and groundwater percentages. We also include yearly land cover dynamics
(i.e., urban, forest, and cropland fraction changes), as well as dynamically changing reservoir
capacity and degree of regulation (DOR) percentage. Leaf area index (LAI) is also included to
reflect the seasonal changes in vegetation canopy that are also key to the streamflow processes.
**Static attributes:** GSHA does not extract updated static attributes because HydroATLAS
already made substantial efforts in this regard. Instead, the listed categories are those mostly related
to streamflow prediction from HydroATLAS selected to be included in GSHA files, and we direct
the readers to the ID match table to access the entire 281 static attributes offered by HydroATLAS
(Lehner et al., 2022; Linke et al., 2019). Our user manual, available at the dataset download site,
also provides more information on it.
**Table 2 Fields provided with GSHA.**

| Category | Field | Description | Unit |
|---|---|---|---|
| Watershed Polygons and basic information | Sttn_Nm | The ID of the watershed. | NaN |
| | Latitude | Latitude of the gauge. | Degree |
| | Longitude | Longitude of the gauge. | Degree |
| | Shedarea | The area of delineated watershed. | $Km^2$ |
| | Agency | The agency the gauge belongs to. | NaN |
| | filename | The name of the corresponding Shapefile in the dataset. | NaN |
| | verification | Verification of watershed area with officially reported area of the corresponding agency. If we did not access the officially reported area of the watershed on the agency website, the field would be "unverified". | NaN |
| | COMID | ID of the MERIT river reach matching with the watershed. | NaN |
| | uparea | Upstream area of the river reach included in the MERIT database. | NaN |
| | order | Stream order of the river reach. | NaN |
| | NextDownID | ID of the downstream river reach in MERIT. | NaN |

| Category | Indices | Description | Unit/Format |
|---|---|---|---|
| Streamflow indices (yearly) | percentiles | Annual 1, 10, 25, 75, 90, 99 percentiles of daily streamflow. | $m^3$/s |
| | mean | Annual mean of daily streamflow. | $m^3$/s |
| | median | Annual median of daily streamflow. | $m^3$/s |
| | annual maximum flood (AMF) | Annual maximum of daily streamflow. | $m^3$/s |
| | AMF occurrence date | The date of AMF occurrence. | Year/month/day |

| | frequency of high-flow events | Number of days in a year with streamflow >= 90 percentile flow. | Days/year |
|---|---|---|---|
| | average duration of high-flow events | Average number of consecutive days >= 90 percentile flow. | Days |
| | frequency of low-flow events | Number of days in a year with streamflow <= 10 percentile flow. | Days/year |
| | average duration of low-flow events | Average number of consecutive days <= 10 percentile flow. | Days |
| | Q=0 days | Number of days with runoff=0. | Days |
| | valid observation days | Number of days with no missing data. (Valid observations refer to non-null measurements.) | Days |

| Category | Indices | Description | Unit/Format |
|---|---|---|---|
| Streamflow indices (monthly) | percentiles | Monthly 1, 10, 25, 75, 90, 99 percentiles of daily streamflow. | $m^3/s$ |
| | mean | Monthly mean of daily streamflow. | $m^3/s$ |
| | median | Monthly median of daily streamflow. | $m^3/s$ |
| | monthly maximum flood (MMF) | Monthly maximum of daily streamflow. | $m^3/s$ |
| | MMF occurrence date | The date of MMF occurrence. | Year/month/day |
| | frequency of high-flow events | Number of days in a month with streamflow >= yearly 90 percentile flow. | Days/month |
| | average duration of high-flow events | Average number of consecutive days in the month >= yearly 90 percentile flow. | Days |
| | frequency of low-flow events | Number of days in a month with streamflow <= yearly 10 percentile flow. | Days/month |
| | average duration of low-flow events | Average number of consecutive days in the month <= yearly 10 percentile flow. | Days |
| | Q=0 days | Number of days with runoff=0. | Days |
| | valid observation days | Number of days with no missing data. | Days |

| Category | Variable | Data source name | Unit |
|---|---|---|---|
| Meteorological Variables | Precipitation | MSWEP | mm |
| | | EM-Earth | mm |
| | 2 m temperature | ERA5 | K |
| | | MERRA-2 | K |
| | | EUSTACE | K |
| | Actual evapotranspiration | REA | mm |
| | | GLEAM | mm |
| | Potential evapotranspiration | GLEAM | mm |
| | | hPET | mm |

| | Radiation (longwave) | ERA5 land surface net thermal radiation | $W/m^2$ |
| | | MERRA-2 surface net downward longwave flux | $W/m^2$ |
| | Radiation (shortwave) | ERA5 land surface net solar radiation | $W/m^2$ |
| | | MERRA-2 surface net downward shortwave flux | $W/m^2$ |
| | 10 m wind speed (u component) | ERA5 land u-component of wind | m/s |
| | | MERRA-2 10 metre eastward wind | m/s |
| | 10 m wind speed (v component) | ERA5 land v-component of wind | m/s |
| | | MERRA-2 10 metre northward wind | m/s |
| | 10 m wind speed (actual) | ERA5 land u- and v-components of wind | m/s |
| | | MERRA-2 10 metre northward and eastward wind | m/s |

| Category | Variable | Data source name | Unit |
| --- | --- | --- | --- |
| Water storage terms | Soil moisture layer 1 | ERA5 land soil water layer 1 (0-7 cm, 0cm refers to the surface) | $m^3/m^3$ |
| | Soil moisture layer 2 | ERA5 land soil water layer 2 (7-28 cm) | $m^3/m^3$ |
| | Soil moisture layer 3 | ERA5 land soil water layer 3 (28-100 cm) | $m^3/m^3$ |
| | Soil moisture layer 4 | ERA5 land soil water layer 4 (100-289 cm) | $m^3/m^3$ |
| | Snow water equivalent | ERA5 land snow depth water equivalent | m of water equivalent |
| | Ground water | GRACE-FO data assimilation | % |

| Category | Variable | Data source name | Unit |
| --- | --- | --- | --- |
| Land use and land cover | Urban fraction | GAUD | % |
| | Forest fraction | MCD12Q1 | % |
| | Cropland fraction | MCD12Q1 | % |
| | Reservoir capacity | GeoDAR | Million $m^3$ |
| | DOR | GeoDAR | % |
| | LAI | CDR LAI | NaN |

| Category | Attribute | Column name (directly from RiverATLAS) | Unit |
| --- | --- | --- | --- |
| Static-Physiography | Elevation | ele_mt_uav | m. a.s.l. |
| | Terrain slope | slp_dg_uav | degrees (x10) |
| | Stream gradient | sgr_dk_rav | decimetres per km |
| Static-Hydrology | Inundation Extent | inu_pc_ult | % |
| | Groundwater Table Depth | gwt_cm_cav | cm |
| Static- | Land Cover Classes | glc_cl_cmj | NaN |

| Landcover | Potential Natural Vegetation Classes | pnv_cl_cmj | NaN |
| | Wetland Extent | wet_pc_u01-u09 | % |
| | Glacier Extent | gla_pc_use | % |
| | Permafrost Extent | prm_pc_use | % |
| Static-Soil & geology | Clay Fraction in Soil | cly_pc_uav | % |
| | Silt Fraction in Soil | slt_pc_uav | % |
| | Sand Fraction in Soil | snd_pc_uav | % |
| | Lithological Classes | lit_cl_cmj | NaN |
| | Soil Erosion | ero_kh_uav | kg/hectare per year |

# 3 Data sources and methodology

## 3.1 Technical workflow in creating GSHA

The creation of GSHA starts from revisiting the data compilation process for the stream
gauging observations from 13 international agencies. The general workflow of GSHA data
production processes is illustrated in **Figure 1**, which consists of watershed delineation, variable
extraction from both grid and non-grid data sources, and uncertainty analysis.
First, we delineated the upstream watersheds using gauge locations. Calibration of gauge
longitudes and latitudes were conducted to match the gauges with the MERIT river network exactly.
The delineated watersheds were selected and manually checked using standards of area, topology
correctness, and observation data lengths. The selected watersheds went on to be overlayed with
grid and non-grid variable data sources for to obtain GSHA variables.

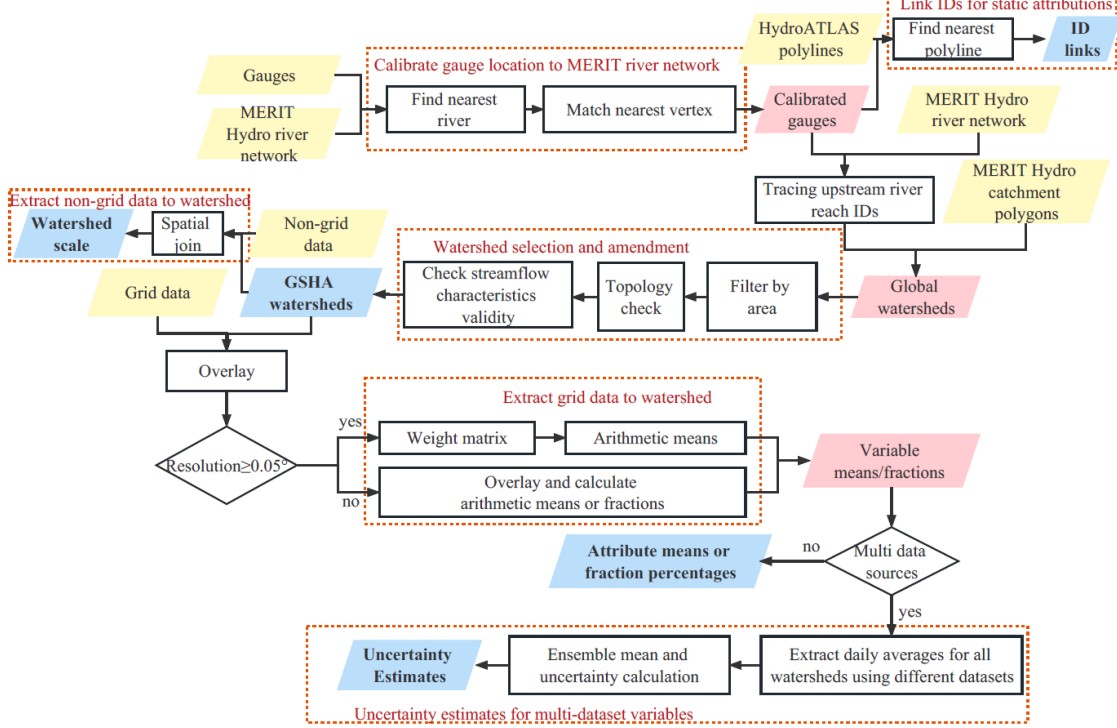


**Figure 1 General workflow of GSHA.** The yellow parallelograms are the input datasets, the blue ones
are the final outputs of GSHA dataset, and the pink ones are the results in the process. The black
quadrilaterals represent the extraction and calculation processes, and the red dotted rectangles illustrate
different modules of the extraction process.
3.2 Gauge-based streamflow indices
As shown in **Table 3**, in total streamflow data from 36497 gauges were initially scraped from
the web and from the Chinese National Real-time Rain and Water Situation Database. For gauges
located within ~100 m of each other, those with fewer years of measurements were removed,
assuming that they are redundant with one another. The gauge measurements were converted to a
consistent unit ($m^3$/s) and then manually compared with GRDC measurements to ensure accurate
unit conversion (Riggs et al., 2023). Gauge databases compiled in this study are available through
a variety of web interfaces, except for the Chinese Hydrology Project (CHP) data which is provided
by the authors of the dataset (Henck et al 2010, Schmidt et al 2011), and processed into annual scale
data that meets the requirements of the synthesis dataset.

**Table 3** Gauge data sources used in this analysis. N1 and N2 refers to numbers of gauges with observations after 1979 and used in GSHA. The starting and ending years (Y1 and Y2) of GSHA gauges for each agency are listed.

| Source | N1 | N2 | Y1 | Y2 | URL/Provider |
|---|---|---|---|---|---|
| ArcticNET 2022 | 116 | 106 | 1979 | 2003 | www.r-arcticnet.sr.unh.edu/v4.0/AllData/index.html |
| Australian Bureau of Meteorology 2022 (BOM) | 4017 | 2340 | 1979 | 2021 | www.bom.gov.au/waterdata/ |
| Brazil National Water Agency 2022 (ANA) | 1343 | 1172 | 1979 | 2021 | www.snirh.gov.br/hidroweb/serieshistoricas |
| Canada National Water Data Archive 2022 (HYDAT) | 3771 | 2222 | 1979 | 2021 | www.canada.ca/en/environment-climate-change/services/water-overview/quantity/monitoring/survey/data-products-services/national- |
| Chile Center for Climate and Resilience Research 2022(CCRR) | 481 | 392 | 1979 | 2020 | https://explorador.cr2.cl/ |
| Chinese Hydrology Project (CHP) | 112 | 26 | 1979 | 1987 | (Henck et al 2010, Schmidt et al 2011) |
| The Global Runoff Data Centre 2022 (GRDC) | 6345 | 4004 | 1979 | 2021 | (https://portal.grdc.bafg.de/applications/public.html?publicuser=PublicU ser |
| India Water Resources Information System 2022 (IWRIS) | 547 | 261 | 1979 | 2020 | https://indiawris.gov.in/wris/#/RiverMonitoring |
| Japanese Water Information System 2022 (MLIT) | 1023 | 751 | 1979 | 2019 | www1.river.go.jp/ |
| Spain Annuario de Aforos, 2022 (AFD) | 1138 | 889 | 1979 | 2018 | http://datos.gob.es/es/catalogo/e00125801-anuario-de-aforos/resource/48368826-e7fd-4a41-950c-89b4eaea0279 |
| Thailand Royal Irrigation Department 2022 (RID) | 126 | 73 | 1980 | 1999 | http://hydro.iis.u-tokyo.ac.jp/GAME-T/GAIN-T/routine/rid-river/disc_d.html |
| U.S. Geological Survey 2022 (USGS) | 16951 | 9069 | 1979 | 2021 | https://waterdata.usgs.gov/nwis/rt |
| Chinese National Real-time Rain and Water Situation Database | 527 | 263 | 2000 | 2019 | http://xxfb.mwr.cn/sq_zdysq.html |

### 3.3 Watershed delineation

The watershed delineation process was built upon a vector-based global river network dataset (Lin et al., 2021), which is delineated from the 90-m Multi-Error-Removed Improved Terrain (MERIT) digital elevation model (DEM) (Yamazaki et al., 2017) and the flow direction and flow accumulation rasters (Yamazaki et al., 2019). The locations of the gauges may contain locational errors and direct delineation will result into erroneous watershed boundaries; therefore, gauge location correction was conducted by relocating the gauges to the nearest MERIT-based river reach vertices. The adjusted gauge points were used as the watershed outlets, where the contributing areas were extracted by dissolving all upstream catchments based on the topology provided by MERIT Basins (Lin et al., 2019). Since the area threshold of MERIT Basins is 25 km$^2$, we did not include watersheds smaller than this threshold. Considering the spatial heterogeneity of very large basins, we excluded watersheds ≥50,000 km$^2$ from the dataset. To ensure GSHA supports studies with sufficiently long records, only watersheds with >5 years of observations since 1979 were selected. For gauges sharing the same watershed, the one with better data quality (i.e., longer measurement records and more valid observation days) was used. If the two gauges share the same quality, we only included the furthest downstream gauge. Eventually, the selection processes resulted in 21568 valid watersheds out of 35970 gauges initially scraped from the web plus 527 gauges from the Chinese National Real-time Rain and Water Situation Database (**Figure 2**).

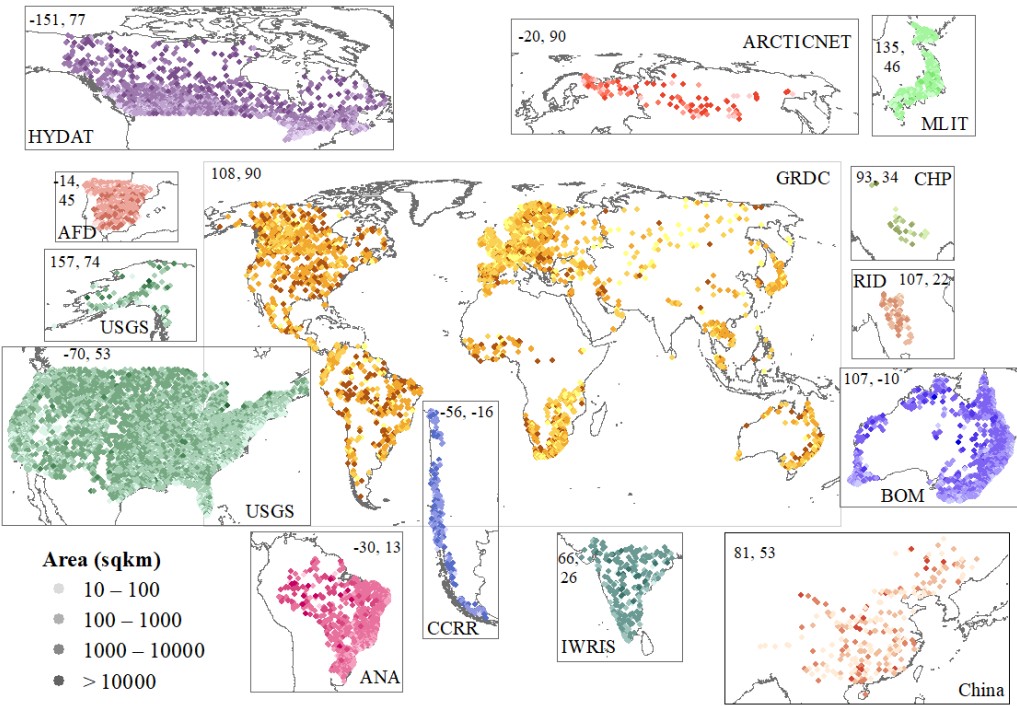

**Figure 2 Spatial distribution of the GSHA gauges (n=21568).** Watershed areas are represented by the tint of colours. Gauges of different agencies are represented with separate colours and are plotted in individual frames (except for USGS gauges in two frames to incorporate Alaska). The agency names and the upper-left coordinates (longitude, latitude) of each frame are also shown in the figure.

The GSHA watersheds are unevenly distributed across the globe, more than half of which are located in North America (USGS, HYDAT, and a large proportion of GRDC gauges, **Figure 3a**).

Europe, Australia, and South America also have relatively good coverage, while Asia and Africa show the lowest gauge densities. The majority of the gauged watersheds are of medium sizes ranging from 250 to 2500 km$^2$, although for some agencies it does not show the same distribution (**Figure 3d**). For instance, ANA (South America), IWRIS (India), and arcticnet (Northern Eurasia) watersheds are generally larger, while the Chinese National Real-time Rain and Water Situation Database provides more gauges with smaller drainage areas. Due to the maintenance difficulties, the number of functioning gauges is declining for agencies like GRDC, but the lack of data in recent years (**Figure 3c**) is mainly due to latency issues. USGS, BOM, and ANA provide a stable number of observations for the 1980-2021 period (**Figure 3c**) with high proportions of valid observations each year (**Figure 3b**), while observational periods from arcticnet and China contain relatively fewer valid samples (**Figure 3b**) and shorter time spans (**Figure 3c**).

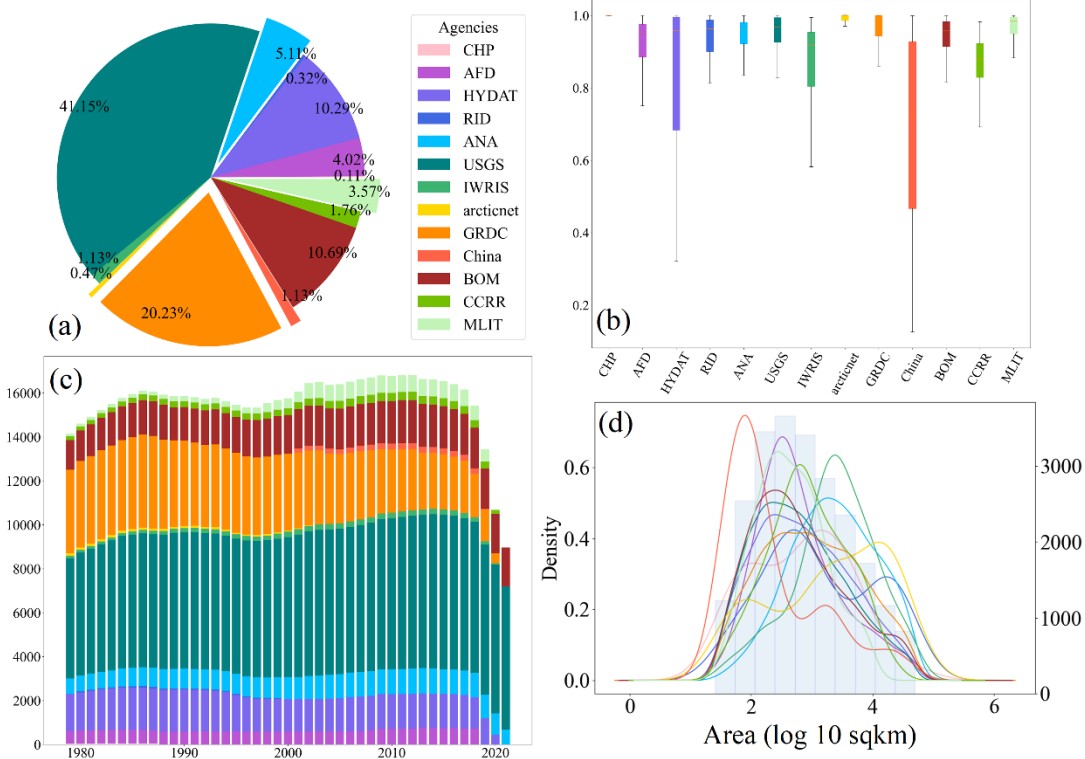

**Figure 3 Summary statistics of the GSHA gauges.** This includes (a) proportions of gauges from different agencies, (b) box plots for proportions of valid observations for each agency, (c) proportion of valid observation for each year by agency and (d) distributions of watershed areas for each agency (kernel density estimation lines, left y-axis) and all gauges (blue histogram, right y-axis). The colour legend in subplot (a) applies to all four subplots. In subfigure (a) the 0.11% label corresponds to CHP, and the legend goes counter clockwise in the pie chart. In subfigure (c), CHP bars are at the bottom of the plot, and the legend goes from bottom to the top of the bars.

## 3.4 Meteorological variables, water storage terms, and land surface characteristics

After watershed delineation, publicly available grid or non-grid data were obtained and overlaid to derive the meteorological, water storage terms, and land surface characteristics. The data sources used for GSHA are listed in **Table 4**. We prioritized the use of multi-source fusion datasets

with relatively high quality surveyed from literature when creating GSHA.
3.4.1 Meteorology datasets
For precipitation, the Multi-Source Weighted-Ensemble Precipitation (MSWEP) that merged
gauge measurements (CPC Unified), grid data (GPCC), satellite products (CMORPH, GSMaP-
MVK, and TMPA 3B42RT), and reanalysis data (ERA-Interim and JRA-55) with sample density
and comparative performance considered (Beck et al., 2017; Beck et al., 2019) are included. Another
precipitation dataset is the Ensemble Meteorological Dataset for Planet Earth (EM-Earth)
deterministic estimates, which merged a station-based Serially Complete Earth (SC-Earth)
removing the temporal discontinuities in raw station observations and ERA5 estimates (Tang et al.,
280 2022).
For 2-m air temperature, the EUSTACE global land station daily air temperature dataset
(EUSTACE) statistically merged station and satellite observations to obtain global daily near-
surface air temperature (Brugnara et al., 2019) is included. Other datasets used for 2-m temperature
extraction are the reanalysis datasets Modern-Era Retrospective analysis for Research and
Applications Version 2 (MERRA-2) (Gelaro et al., 2017) and the fifth generation of European
Reanalysis (ERA5) dataset land component (Muñoz-Sabater et al., 2021). MERRA-2, produced
by NASA's Global Modelling and Assimilation Office (GMAO), used the Goddard Earth Observing
System (GEOS) model and analysis scheme and assimilated the latest observations. ERA5
reanalysis was developed by the European Centre for Medium-Range Weather Forecasts (ECMWF)
using the Carbon Hydrology-Tiled ECMWF Scheme for Surface Exchanges over Land
(CHTESSEL) driven by the downscaled meteorological forcing from the ERA5 climate reanalysis
(Hersbach et al., 2020). These reanalysis datasets are also used in extracting long- and shortwave
radiation, as well as u- and v-components of wind.
For AET, the REA dataset, which used the reliability ensemble averaging (REA) method to
merge ERA5, Global Land Data Assimilation System Version 2 (GLDAS2), and MERRA-2 is used
(Lu et al., 2021). Another AET data source is the product of the Global Land Evaporation
Amsterdam Model (GLEAM) based on satellite observations of surface net radiation and near-
surface air temperature (Martens et al., 2017). For PET, GLEAM is also incorporated. Another PET
dataset for GSHA is an hourly PET at 0.1° resolution for the global land surface (hPET) calculated
from ERA5-land wind speed, air and dew point temperature, net radiation components, and surface
air pressure (Singer et al., 2021).
3.4.2 Water storage term datasets
ERA5-land data is also applied in extracting soil moisture for 4 soil layers, as well as snow
water equivalence. For groundwater, an assimilation dataset from NASA's Gravity Recovery and
Climate Experiment (GRACE) and its follow-on mission (GRACE-FO) is used (Li et al., 2019).
The dataset merged water storage derived from GRACE satellite products into ECMWF Integrated
Forecasting System meteorological data-forced NASA's Catchment land surface model (CLSM).
The data is represented as groundwater drought indicator (GWI), which is the percentage of
groundwater storage estimates from the GRACE data assimilation relative to the climatology
(representing historical conditions), at weekly time scales from 2003-2021.
3.4.3 Land surface characteristic datasets
Global urban development for 1985-2015 is represented as the urban fraction in each watershed
using the global annual urban dynamics (GAUD) at 30-m resolution. The dataset was derived from
Landsat surface reflectance based on the Normalized Urban Areas Composite Index (NUACI) (Liu
et al., 2020). For forest and cropland fractions, the Terra and Aqua combined Moderate Resolution
Imaging Spectroradiometer (MODIS) Land Cover Type (MCD12Q1) land cover dataset, is used
(Friedl et al., 2010). It covers 2001-2020 with a resolution of 500 m, and the categories used for
GSHA are the International Geosphere–Biosphere Programme classification (IGBP) forests and
croplands. Another land cover is vegetation, which is represented by LAI obtained from the National
Oceanic and Atmospheric Administration (NOAA) Climate Data Record (CDR) of Advanced Very
High-Resolution Radiometer (AVHRR) product, which relied on artificial neural networks and
AVH09C1 surface reflectance product (Claverie et al., 2016).
3.4.4 Dams and reservoirs
The newly published Georeferenced global Dams And Reservoirs (GeoDAR) dataset that
documented the dam and reservoir construction years is used for building the temporally varying
watershed reservoir capacity and DOR. GeoDAR georeferenced the International Commission on
Large Dams (ICOLD) World Register of Dams (WRD), and geo-matched multi-source regional
registers and geocoding descriptive attributes through the Google Maps API (Wang et al., 2022).
The reservoir capacities are used together with the mean annual streamflow to obtain the DOR based
on equation $dor = SC/Q_{mean}$, where $SC$ refers to reservoir storage capacity and $Q_{mean}$ is the
mean annual streamflow in the corresponding year.
3.4.5 Static variables
We matched GSHA river IDs and HydroATLAS river reach IDs to link the static attributes.
HydroATLAS includes 56 variables for hydrology, physiography, climate, land cover & use, soils
& geology, and anthropogenic influences for over 8.5 million river reaches globally.
**Table 4 Data sources used for the GSHA variables.**

| Category | Dataset | Resolution | Interval | Reference |
|---|---|---|---|---|
| Meteorology | MSWEP | 0.25° | Daily | (Beck et al., 2017; Beck et al., 2019) |
| | EM-Earth | 0.1° | Daily | (Tang et al., 2022) |
| | ERA5-land | 0.1° | Hourly | (Muñoz-Sabater, 2019) |
| | MERRA-2 | 0.5°* 0.625° | Hourly | (GMAO, 2015) |
| | EUSTACE | 0.25° | Daily | (Brugnara et al., 2019) |
| | REA | 0.25° | Daily | (Lu et al., 2021) |

| | | | | |
|---|---|---|---|---|
| | GLEAM | 0.25° | Daily | (Martens et al., 2017; Miralles et al., 2011) |
| | hPET | 0.1° | Daily | (Singer et al., 2021) |
| Water storage terms | ERA5-land | 0.1° | Hourly | (Muñoz-Sabater, 2019) |
| | GRACE-FO data assimilation | 0.25° | Weekly | (Li et al., 2019; Zaitchik et al., 2008) |
| Land surface | GAUD | 30 m | Yearly | (Huang, 2020) |
| | MCD12Q1 | 500 m | Yearly | (Friedl et al., 2019) |
| | CDR Leaf Area Index | 0.05° | Daily | (Vermote et al., 2019) |
| Dam and reservoir | GeoDAR | NaN (polygon) | Yearly | (Wang et al., 2022) |
| Static Attributes | HydroATLAS | NaN (line) | NaN (static) | (Lehner et al., 2022; Linke et al., 2019) |

## 3.5 Variable extraction methods

For grid data with relatively coarse spatial resolutions (≥0.05°), we used an area-weighted approach to extract the variable (Addor et al., 2017) based on the proportion of the grid area contained in the basin boundary, while for high-resolution grid data, we extracted the arithmetic mean directly. **Figure 4** shows the area-weighted average approach we used for grid data with spatial resolution ≥0.05° to reduce the influence of watershed area on data uncertainty (Tang et al., 2022). The grid data (**4a**) and the quality-controlled watersheds (**4b**) were overlayed and all grids intersecting with the watershed were obtained (**4c**). For each intersected grid, the proportion of the polygon in the grid was calculated as the weight (dark blue, **4d**); the product of the weight and the corresponding grid value was calculated over all intersected grids (**4e**) and were summed up as the weighted average (**4f**). For wind, the u- and v-wind components were first used to calculate wind speed, then the basin average was calculated with the weighted average approach. For grid data with a spatial resolution of <0.05°, the area-weighted approach was not adopted as it offers limited gains while becoming computationally too expensive. For reservoirs, we used the reservoir polygons in GeoDAR, which were spatially joined to GSHA watershed polygons. All the intersected reservoirs were considered contributory to the management of the corresponding watershed and were used to calculate the total reservoir storage capacity and degree of regulation.

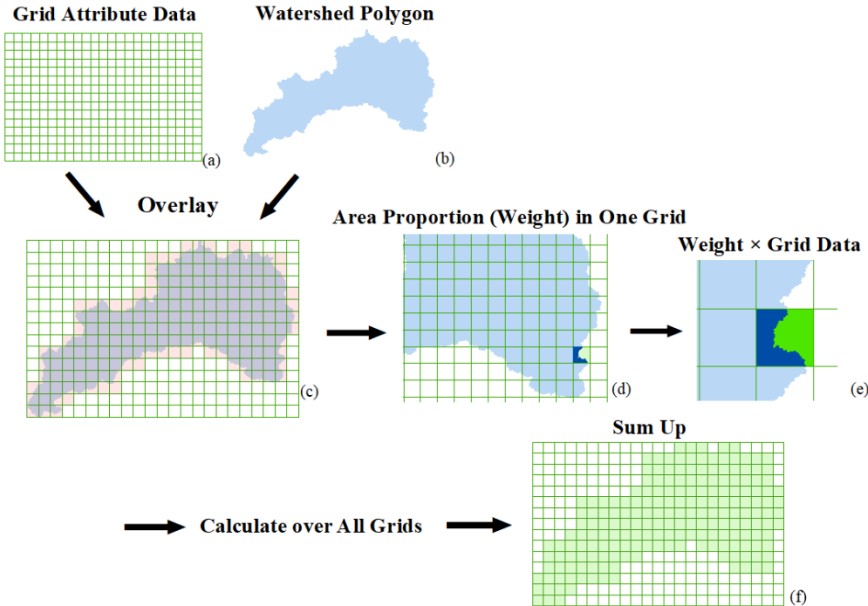


**Figure 4 Determination of the area weights in extracting gridded data to GSHA watershed**
**polygons.** This weighted approach is applied to data at a resolution of ≥0.05° but not for data at a finer
spatial resolution due to computational costs.
3.6 Uncertainty estimates
We also provided uncertainty estimates of the meteorological variables by calculating the long-
term mean of each dataset in each watershed, where the discrepancy between the maximum and
minimum among the data sources ($X_{max}$ and $X_{min}$) as a percentage of their mean ($\bar{X}$) was used
in the uncertainty estimation (see Eq. 1):

$$uncertainty = \frac{X_{max}-X_{min}}{\bar{X}} * 100\%, \qquad (1)$$


3.7 Validation
After delineation, we validated our watershed areas with officially reported watershed areas
from BOM, HYDAT, and GRDC by matching GSHA watersheds by their agency IDs. We set the
criteria of mismatched watersheds as (1) the area difference being over $\pm20\%$ of the officially
reported area, and (2) the area ratio being less than 0.1 or over 10 times the reported areas. Since
not all agency websites reported watershed areas, thus we added a flag field in the attributes as
"unverified", "verified match", and "verified mismatch" to allow users to filter the watersheds
flexibly and avoid putting the samples in the dataset under an unfair standard.
Postprocessing of the extracted variables includes the unification of units and manual quality
checks. For streamflow characteristics, we validated three of our indices against GSIM for its global
coverage, including the mean annual streamflow, 10th and 90th percentiles. The spatial joint between
GSHA and GSIM gauges in a 10 km buffer zone was performed, and only the GSIM gauge with a
minimum distance and watershed area difference ≤5% to a GSHA gauge was considered. Pairs
with 0 measurements were excluded and 9835 pairs were involved eventually. We plotted the scatter
plot of GSHA-GSIM mean flow, 10-th and 90-th percentiles, and compared the fitting line to the 1:1
line, with correlation coefficients calculated (see Section 4.1).
We also validated precipitation, potential ET, and 2 m air temperature with the regional
CAMELS-US dataset. We compared the Daymet meteorological variables of CAMELS and the
mean of GSHA variables for validation. Since we included ERA5 data for most of our variables
directly or indirectly as the data source, while Caravan consistently used ERA5, we did not use
Caravan for the global validation as it is not considered as fully independent from GSHA. The
spatial match was the same as we did for GSIM which resulted in 906 pairs. This number was larger
than the total CAMELS gauge numbers as some gauges might be repeatedly paired due to location
bias of the USGS gauges and MERIT river networks, as well as the adjacency between gauges of
different agencies. Similarly, scatter plots and correlation coefficients are provided for assessment.
3.8 Watershed classification and change detection
We classified the watersheds as natural and human-managed to analyse the influence of human
water management. A watershed is classified as a natural watershed if it satisfies the following: (1)
DOR is smaller than 10%; (2) the urban extent is less than 5%; and (3) the sum of urban and cropland
fractions is smaller than 10% (L. Yang et al., 2021; Zhang et al., 2023). The classification was
performed for 2001-2015, and the changing patterns of the watersheds are divided into six categories:
(1) natural (N) when the watershed remained natural for all 15 years; (2) human managed (H) when
the watershed remained human managed for all 15 years; (3) natural to human managed (NH) when
the watershed was first natural in 2001, but changed to and maintained human managed later; and
(4) human managed to natural (HN) when the watershed was first human managed in 2001, but
changed to and maintained natural later.
# 4 Results
As previous studies have already revealed the spatial patterns of the LSH hydrometeorological
variables both locally and globally, here we put the spatial patterns of GSHA meteorological
variables and streamflow indices in **Appendix A**, while we focus on using the Results section to
reveal the uniqueness of GSHA. These include a technical validation of GSHA, uncertainty analysis,
and the temporal change of watershed human management levels.
4.1 Technical validation
The validation result figures of watershed areas are in **Appendix B** since we focused more on
the variables and already added the validity results in the dataset as "unverified", "verified match",
and "verified mismatch" fields in the dataset. Under our criterion of filtering "mismatch" watersheds,
1.9% of BOM watersheds, 4.7% of HYDAT watersheds and 8.9% of GRDC watersheds are
mismatched. After removing these watersheds, correlation coefficients between GSHA and the
agencies can reach 0.99, which verified the correctness of our watershed delineation and data
extraction approach.

**Figure 5** illustrates the validation results of GSHA. **Figures 5a–5c** show streamflow indices

as validated against GSIM globally, and **Figures 5d–5f** show meteorological variables as validated
against Daymet from CONUS CAMELS. For streamflow indices, precipitation, and temperature,
the correlation coefficients exceed 0.95 (significance p<0.01), and the fitting lines are close to the
1:1 line, indicating high consistencies between GSHA and the reference datasets. For PET, however,
the coefficient is low, at only 0.573 (significance p<0.05), and the CAMELS PET is generally higher
than GSHA ensemble, which is possibly ascribed to the high uncertainty among PET datasets that
is yet to be fully resolved (Singer et al., 2021) (see **Appendix C**). Note that the gauge pairing might
bring a small proportion of wrong pairs for some very close gauges, and differences in temporal
ranges of GSHA and GSIM might cause some discrepancies for observed streamflow.

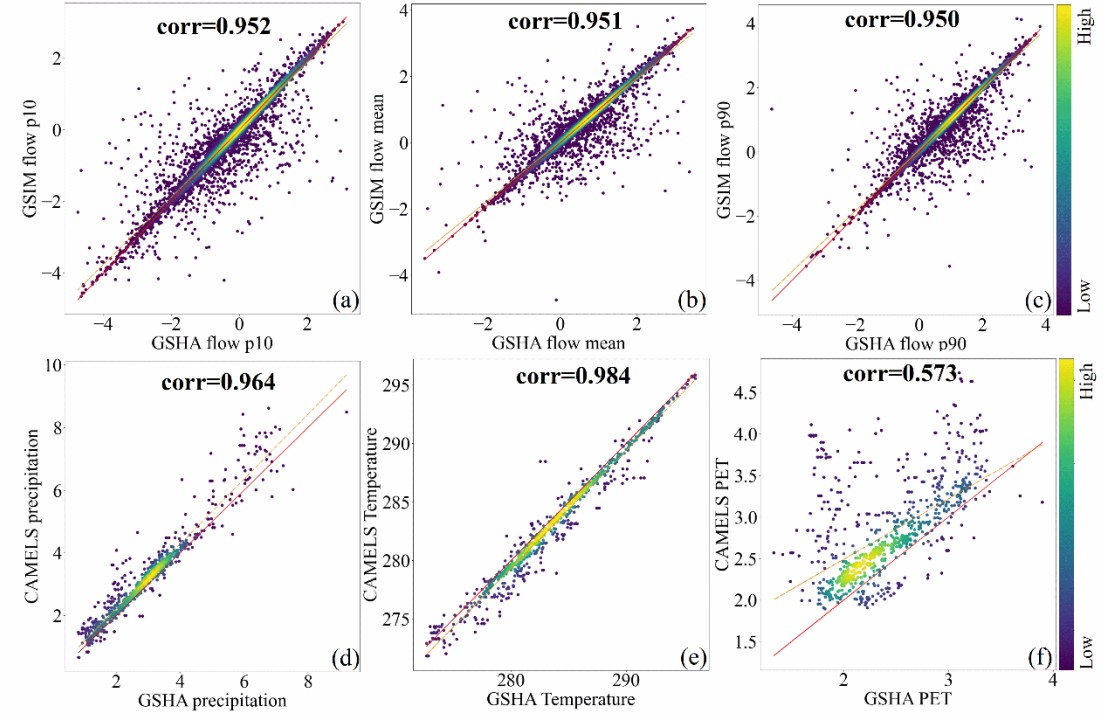


**Figure 5 Validation of GSHA with GSIM streamflow characteristics ((a), (b) and (c)), and**
**CAMELS meteorological variables ((d), (e) and (f)).** 'Corr' in the subfigure is the Pearson correlation
coefficient. The red line is the 1:1 line, while the orange dotted line is the fitting line of the scatter points.
The colour bar represents density of the sample points. The unit of X and Y axes in (a), (b). and (c) is
long10 m³/s.
4.2 Uncertainty patterns for the GSHA meteorological variables

**Figure 6** shows the distributions of the uncertainties for different variables, and the colour bars

are unified to allow for comparisons between different variables.

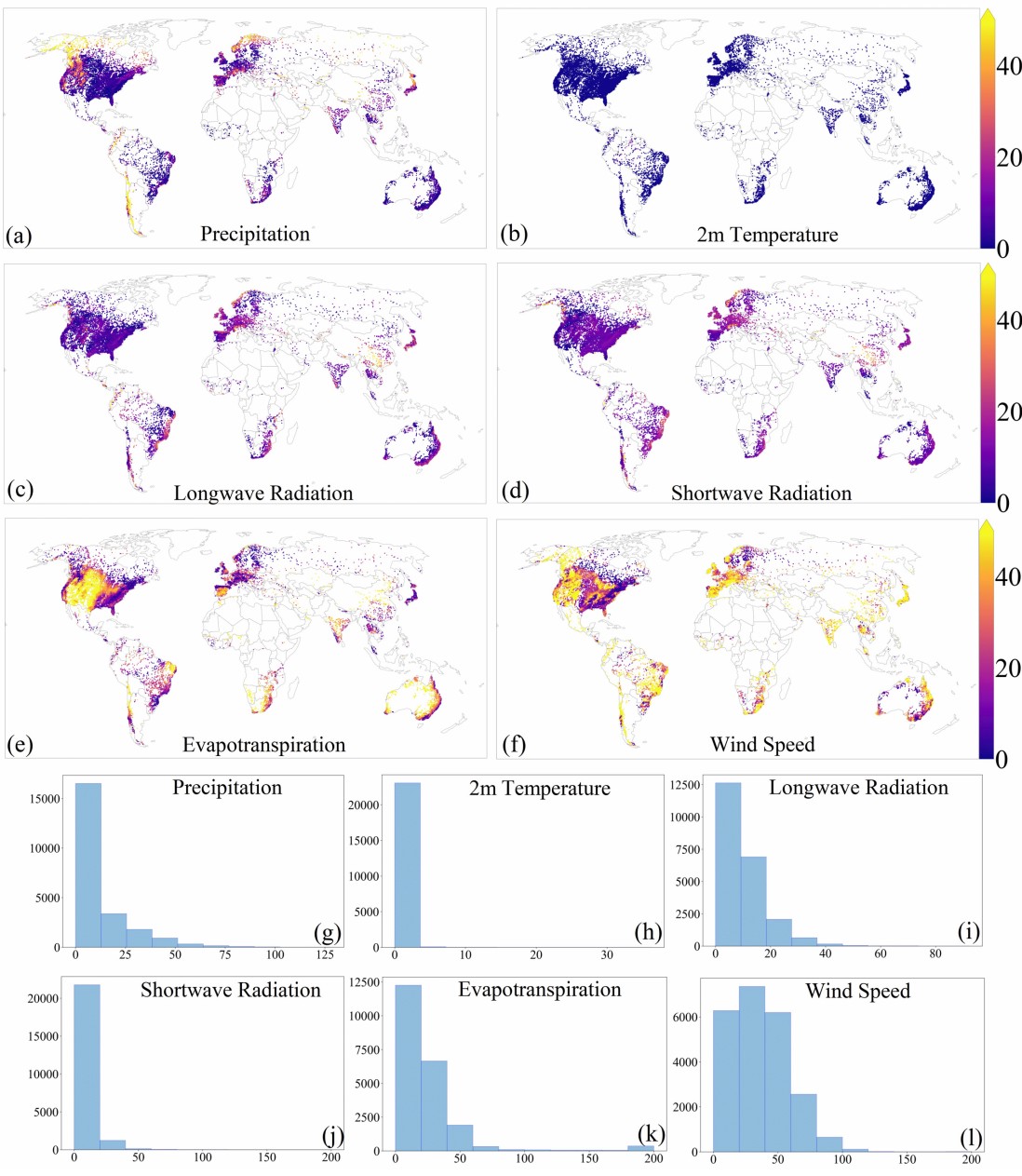

**Figure 6 Global patterns of the uncertainty for the GSHA meteorological variables (in percentage).** This includes the uncertainty (a) for precipitation (mm/day), (b) 2-m temperature (K), (c) longwave radiation (W/m$^2$), (d) shortwave radiation (W/m$^2$), (e) evapotranspiration (mm/day), and (f) wind speed (m/s), and (g) the uncertainty histogram for precipitation, (h) 2-m temperature, (i) longwave radiation, (j) shortwave radiation, (k) evapotranspiration, and (l) wind speed.

Generally, among all variables, air temperature (**Figures 6b & 6h**) shows the minimum uncertainty (<5%), suggesting high consistency of air temperature estimates from different datasets. The uncertainty for wind speed (**Figure 6f**) is the highest among all variables. Uncertainties for other variables show strong spatial variability. For example, uncertainties for precipitation are high in high-latitude or mountainous areas like the Rocky Mountains, northern Europe, the Alps, and the Andes areas (**Figure 6a**). This is reasonable because limited accessibility to in-situ observations and the misestimation of snow (Schreiner-McGraw & Ajami, 2020) can contribute to precipitation

estimation errors, while the data sources show relatively high consistency (*uncertainty* ≤25%) in other parts of the world (**Figure 6g**). For radiation, as solar/shortwave radiation is largely affected by sky conditions, uncertainties are high in regions with less clear sky, including south-west China and its surrounding areas, high latitude regions of the northern hemisphere, and Europe (Brun et al., 2022). These places are also subject to high thermal/longwave radiation uncertainties for similar reasons (**Figure 6c**). Land cover including vegetation and artificial surface, is another factor influencing surface net radiation through the albedo effect (Hu et al., 2017), thus for heavily vegetated and urbanized areas, such as the Amazon region and east coastal Australia, uncertainties for both longwave and shortwave fluxes are also relatively high. Nevertheless, **Figures 6i & 6j** demonstrate that for the majority of watersheds, radiation uncertainties are < 25%, indicating that the radiation data sources are generally consistent with each other. ET uncertainties are generally larger than the above variables (**Figures 6e & 6k**), and are particularly prominent in dry areas of the globe, e.g., central North America, northern Andes, central Asia, and Australia's grasslands and deserts. It is also prominent in agriculture intensive regions like India and the northern part of China (Sörensson & Ruscica, 2018), where agricultural irrigation may be the contributing factor to the ET uncertainty. The spatial distributions of wind speed do not seem to show clear regional patterns (**Figure 6f**), and uncertainty values of wind speed are generally larger over the majority of watersheds (**Figure 6l**). Nevertheless, the uncertainties are low in Appalachia and northern Europe, and are high in most parts of Brazil, the Andes, Africa, eastern and southern parts of Asia, as well as Australia (**Figure 6f**). As we already selected relatively high-quality datasets for the variables, these areas might be calling for more attention by the LSH developers, while providing possible explanations for the inconsistencies in interpreting results or understanding the challenges in estimating model parameters by the LSH users.

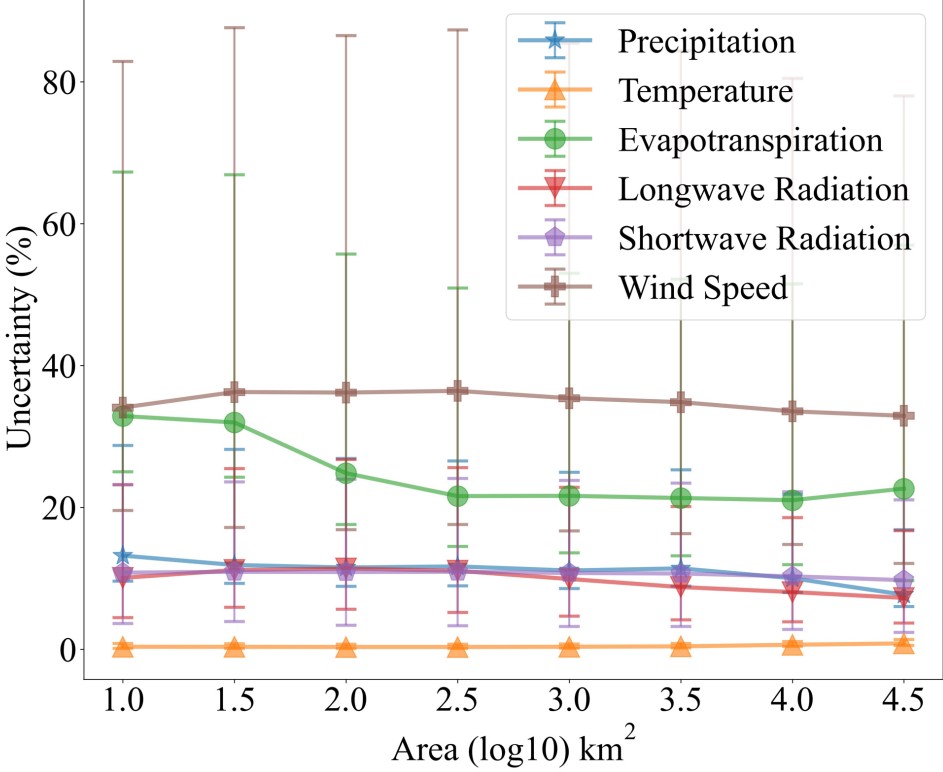

**Figure 7 Relationship between variable uncertainties and watershed areas.** The markers indicate mean values of the variable uncertainties in watersheds smaller than the corresponding x-axis value. The error bars represent the range between 25 and 75 percentiles of the uncertainty values.

Apart from the spatial patterns above, we also investigated the emergent patterns of the uncertainties. Existing studies indicate small basins can show larger uncertainties due to coarse resolution data inputs (Kauffeldt et al., 2013), while sub-grid variabilities might be offset by averaging over large watersheds. As we plotted the uncertainty against watershed areas in **Figure 7**, it verifies that for most variables, the uncertainty declines as the watershed area increases. **Figure 7** also reveals some interesting patterns which were rarely discussed in existing studies. For example, the most obvious decline of data uncertainty with area came from ET (green). ET is highly dependent on and significantly affected by land surface spatial heterogeneity, thus it benefits the most from spatial averaging for large river basins. Longwave radiation uncertainty (red) experiences a moderate decline, likely due to its linkage with land surface complexity and cloud conditions. Shortwave radiation and precipitation uncertainty show a similar decline pattern (blue and purple), which is possibly related to their strong ties to cloud covers. Temperature has a low uncertainty, and its relationship to watershed area is also not obvious. Wind speed uncertainty only declines slightly as the area increases, and this may be because wind speed uncertainty can be traced back more to the atmospheric circulation patterns instead of land surface conditions, thus showing a non-prominent relationship with watershed area. Overall, GSHA provides uncertainty estimates that capture these prominent patterns, which can be helpful to hydrologic modellers and users.

4.3 Natural and human managed watersheds and changing patterns

We also demonstrate the other key features of GSHA by categorizing global watersheds into
natural and human-managed, and more prominently their temporal shifts in **Figure 8**. Overall, the
majority of human-managed watersheds are located in the US, Europe, and other regions with
intensive industrial or agricultural activities such as East and South Asia (**Figures 8a and 8b**).
During 2001-2015, 46.89% of the watersheds remained natural, while another 47.62% under human
management in 2001 remained in the category throughout the study period (**Figure 8d**). Generally,
the northern hemisphere has a larger proportion of human-managed watersheds, while watersheds
in the less populated and urbanized southern hemisphere largely remain natural.

Noticeably, 4.36% of GSHA watersheds switched from natural to human-managed (1011
watersheds), and the remaining 1.13% changed back to natural states from human managed during
2001-2015. For instance, watersheds in the middle and lower Yangtze River area and the north-
eastern China show a shift from human-managed to natural state, where ecological restoration
projects were in place (Qu et al., 2018; Zhang et al., 2015). Although the time span of GSHA LULC
dynamics restricted the change detection for developed countries as their urbanizations and
infrastructure developments have long been completed, and for fast emerging economies after 2015,
the time series were also missing; nevertheless, the changing human activities captured by GSHA
may be helpful to understand the streamflow changes including flood characteristics (Yang et al.,
2021; Zhang et al., 2022).

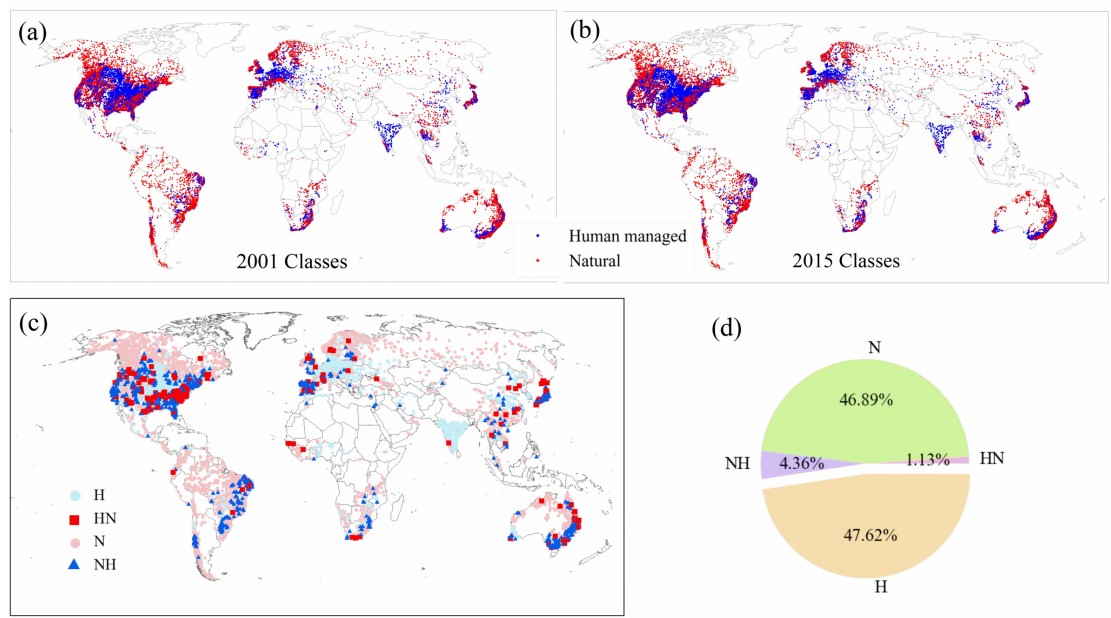


**Figure 8 Classification of natural and human managed watersheds in 2001 (a) and 2015 (b).**
**Changes in watershed categories are illustrated by (c) and (d).** H and N in (c) and (d) represent
watersheds that maintained human managed or natural from 2001-2015; NH and HN represent those
changing from natural to human managed and from human managed to natural, respectively.

We further used several examples to illustrate the changing status of GSHA watersheds (**Figure**
**9**). **Figures 9a and 9b** show a watershed located in Northeast China, where the rapid increase in

cropland shifted the watershed from natural states to human-managed in recent years. **Figures 9c and 9d** correspond to a mountainous area in Sichuan Province, China, which became human-managed due to the construction of a reservoir in 2006. For another case in Northeast China (**Figures 9e and 9f**) and a USGS case (**Figures 9g and 9h**), the watersheds shifted from human-managed to natural, which is mainly manifested by the reduction in cropland fraction due to the environmental policy. For instance, afforestation during 2000-2010 in Changbai Mountains where the watershed in **Figures 9e and 9f** is located, significantly increased the forest cover and might bring a decline in human disturbance in the form of land use (Zhang & Liang, 2014). These results highlight the shifting watershed status that would require further attention from LSH users, which is encapsulated in GSHA v1.0 and will be continuously improved in the future.

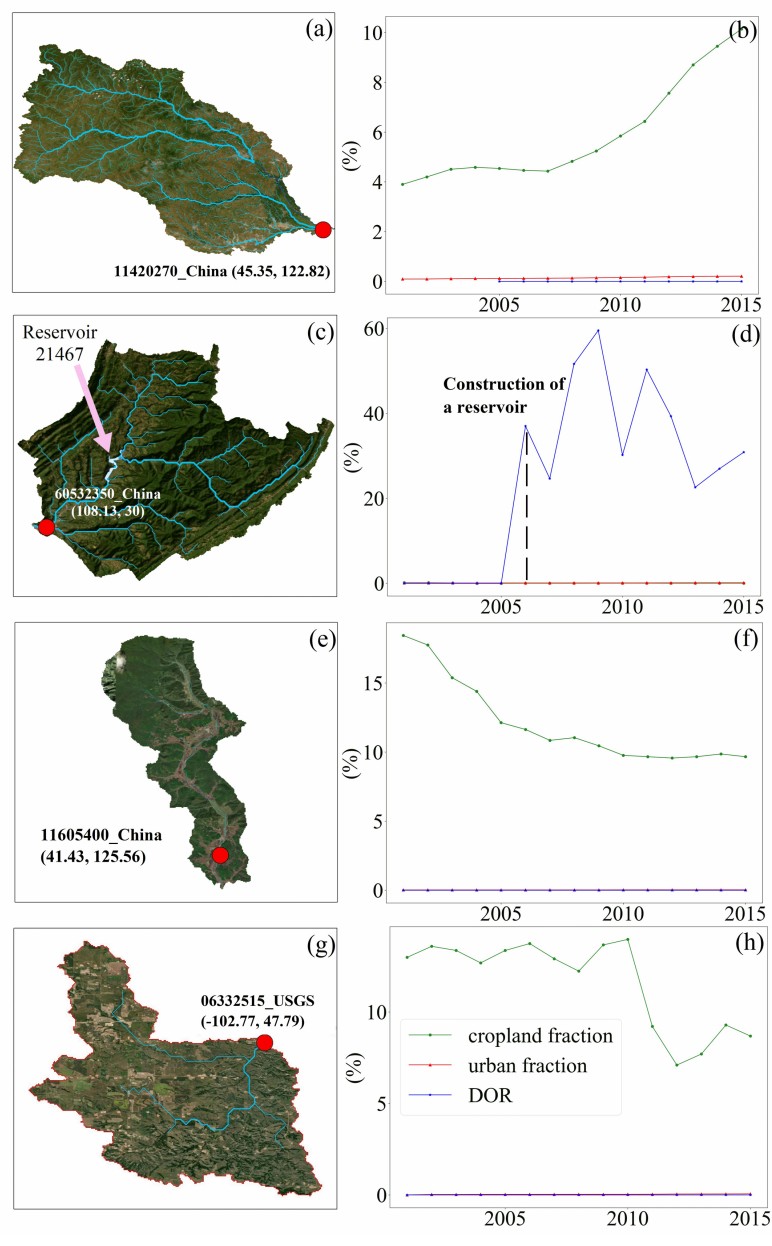

**Figure 9 Cases for shifting status of the watershed classification.** (a) and (b) correspond to 11420270_China, and (c) and (d) correspond to 60532350_China, both of which changed from natural to human managed category. (e) and (f) represent11605400_China, and (g) and (h) correspond to

06332515_USGS watershed changing from human managed to natural watershed.

## 4.4 Changing runoff coefficient patterns derived from GSHA

Finally, we also analysed the global pattern in the trend of runoff coefficient (RC) as a brief
demonstration on what GSHA can offer out of its many potential usages. RC is defined as $R/P$,
where R denotes runoff (mm) and P denotes precipitation (mm). **Figure 10a** shows that regions with
high RC (i.e., a large proportion of rainfall goes into rivers instead of being evaporated or consumed)
are in east Asia and North America, most parts of Europe, the west coast of North America and the
Amazon, in general agreement with the aridity patterns across the globe. For arid/semiarid areas
and places with intense water use (e.g., western US, eastern Brazil, Australia, Africa), RC is low,
meaning most of the precipitation does not reach the gauged river.
We found that RC generally remained stable for the past decades (i.e., grey dots in **Figure**
**10b**; >80% of the gauges did not observe a statistically significant trend), while 4252 watersheds
observed a statistically significant trend in RC at 95% level (5690 watersheds at 90% level). Among
them, decreasing RC is more widespread than increasing RC. The most pronounced decreasing
trends are observed in Europe, India, eastern Brazil, Chile, eastern Australia, and the Euphrates and
Tigris, which largely correspond to regions with known intense agricultural, industrial, and
residential water use that may have reduced the river water. We note that the global RC trend patterns
were different from a recent study that showed mostly increasing RC in the high-latitudes, central
North America, eastern Australia, and Europe (Xiong et al., 2022). Given Xiong et al. (2022) used
estimated runoff while we used runoff directly from gauge observations, it is likely that the
concerning water availability issues in the context of increasing human water use may not be fully
captured by existing studies. Regional studies also tend to show inconsistent results. For example,
a study based on models incorporating climate change and land use change but ignoring human
water consumptions suggested that deforestation and urbanization generally increase RC (Lucas-
Borja et al., 2020), while another study identified a significant decreasing trend for RC by focusing
on cases with intense irrigational water use (Banasik and Hejduk, 2012). These collectively preclude
a clear identification of consistent RC trends (Velpuri and Senay, 2013) and a clear causal factor
attribution given the complexity of the anthropogenic factors. As such, GSHA may offer a new path
to fill in the gap of disentangling the influences of large-scale water use on decreasing RC.

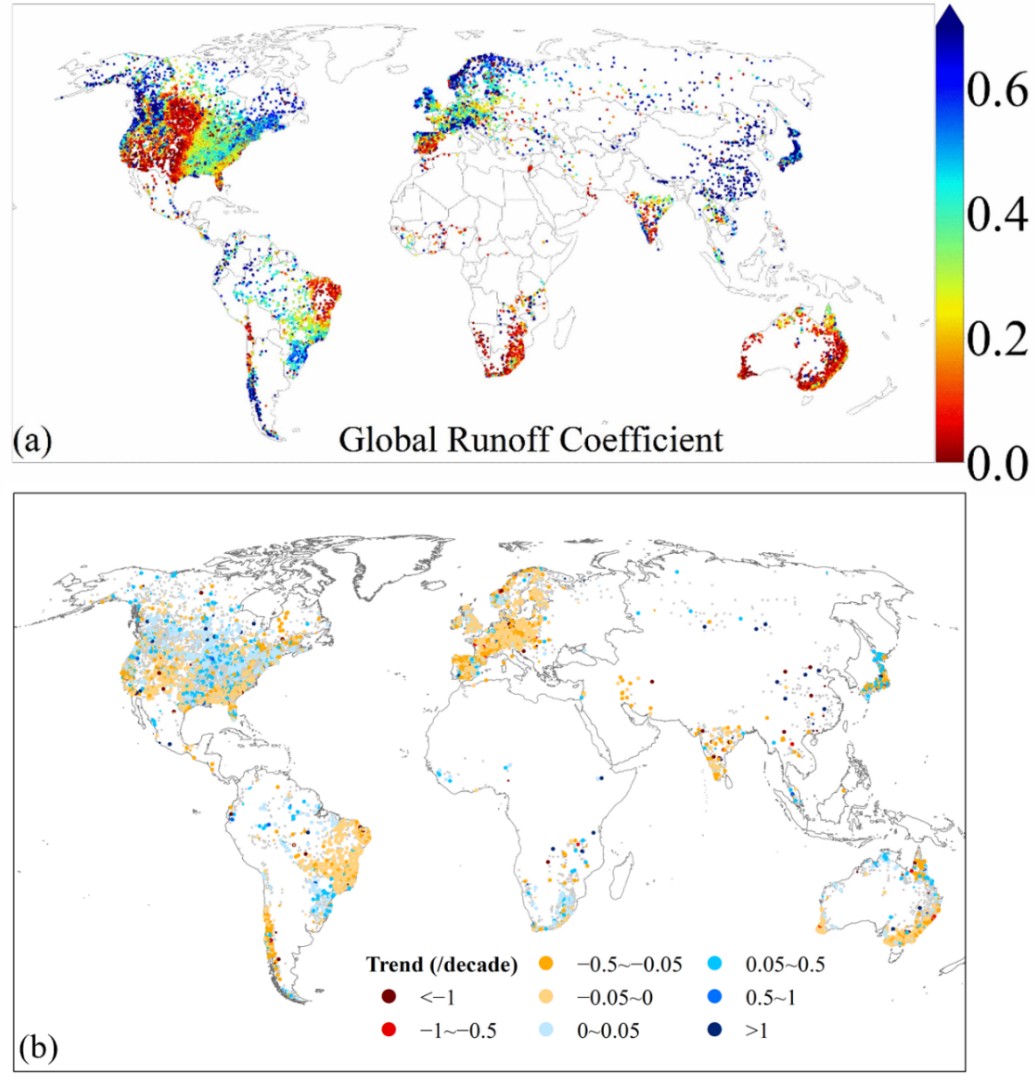

**Figure 10 Patterns of runoff coefficient (a) and its trend (b).** Only watersheds with statistically significant trend (p<0.05) are shown with colours in (b); the small and large sized points represent 95% (p<0.05) and 90% significance level (p<0.1), respectively. Note that the temporal coverage is different for different gauges; readers can refer to the GSHA temporal coverage for interpreting the patterns. The figure illustrates 18987 GSHA watersheds. Watersheds with less than 10 years of indices calculated from over 250 valid observations per year, as well as with runoff coefficient trend over 20 per decade, are not demonstrated in subfigure b.

# 5 Conclusions

Large sample hydrology (LSH) datasets play a critical role in data-driven analyses and model parameter estimation for hydrological studies. From MOPEX (Duan et al., 2006) to Caravan (Kratzert et al., 2023), significant efforts have been made to improve the comprehensiveness of LSH, yet issues related to data spatial coverage, uncertainty estimates, and human activity dynamics remain to be solved. This study complements existing LSH with a new synthesis dataset named the

Global Streamflow characteristics, Hydrometeorology, and catchment Attributes for large sample
river-centric studies (GSHA v1.1).
To summarize, GSHA contributes the following aspects to the LSH development:
1.  It includes streamflow indices, hydrometeorological data, and surface characteristics data for
21568 gauges compiled from 13 agencies worldwide, which represents one of the most
comprehensive LSH by far.
2.  We incorporated multiple data sources to provide uncertainty estimates for each meteorological
variable (including precipitation, 2 m air temperature, radiation, wind, and ET). The spatial
patterns and the relationship between the uncertainty and the watershed characteristics GSHA
reveals may be helpful to identify inconsistencies among data-driven studies or biases for model
parameter estimation studies using existing LSH.
3.  Dynamic data are provided for previously static data descriptors for land cover changes
including urban, cropland and forest fractions, as well as reservoir storage change including
storage capacity and degree of regulation.
Although GSHA does not cover watersheds of $<25km^2$ or the dynamics of cryosphere variables
(e.g., glacier and permafrost) that have become increasingly important in terrestrial hydrological
changes, and the time spans for the dynamic descriptors of LULC are unable to cover the critical
periods for the advanced and less-advanced economies due to the constraints with existing LULC
data, GSHA is expected to be utilized to unravel the following insights:
1.  The uncertainty patterns vary between variables and geographical regions, indicating that the
interpretation of model and analysis results need to consider inconsistencies of raw data, apart
from looking into the methodologies and patterns themselves.
2.  Although most watersheds have remained natural or human managed throughout the GSHA
time span, a considerable number of watersheds shifted between the two categories, which can
be ascribed to urbanization, cropland increase, reservoir construction and ecological restoration
such as returning farmland to natural states, and these can be clearly manifested using GSHA.
3.  Analysis with runoff coefficient reveals that among gauges with a statistically significant trend,
a greater portion experienced a declining RC trend than an increase trend. This pattern revealed
by GSHA can be used to further study water availability issues in a changing climate.
As our knowledge on the above processes continues to improve, we expect that future versions
of GSHA will be continuously updated. Finally, better hydrological data sharing is crucial to
advance global change hydrology studies.
# Appendix
**A. Spatial patterns of GSHA meteorological variables**
**Figures A1 & A2** show the spatial distributions of GSHA meteorological variables and selected
streamflow indices. The spatial pattern derived from each individual data source is plotted separately.

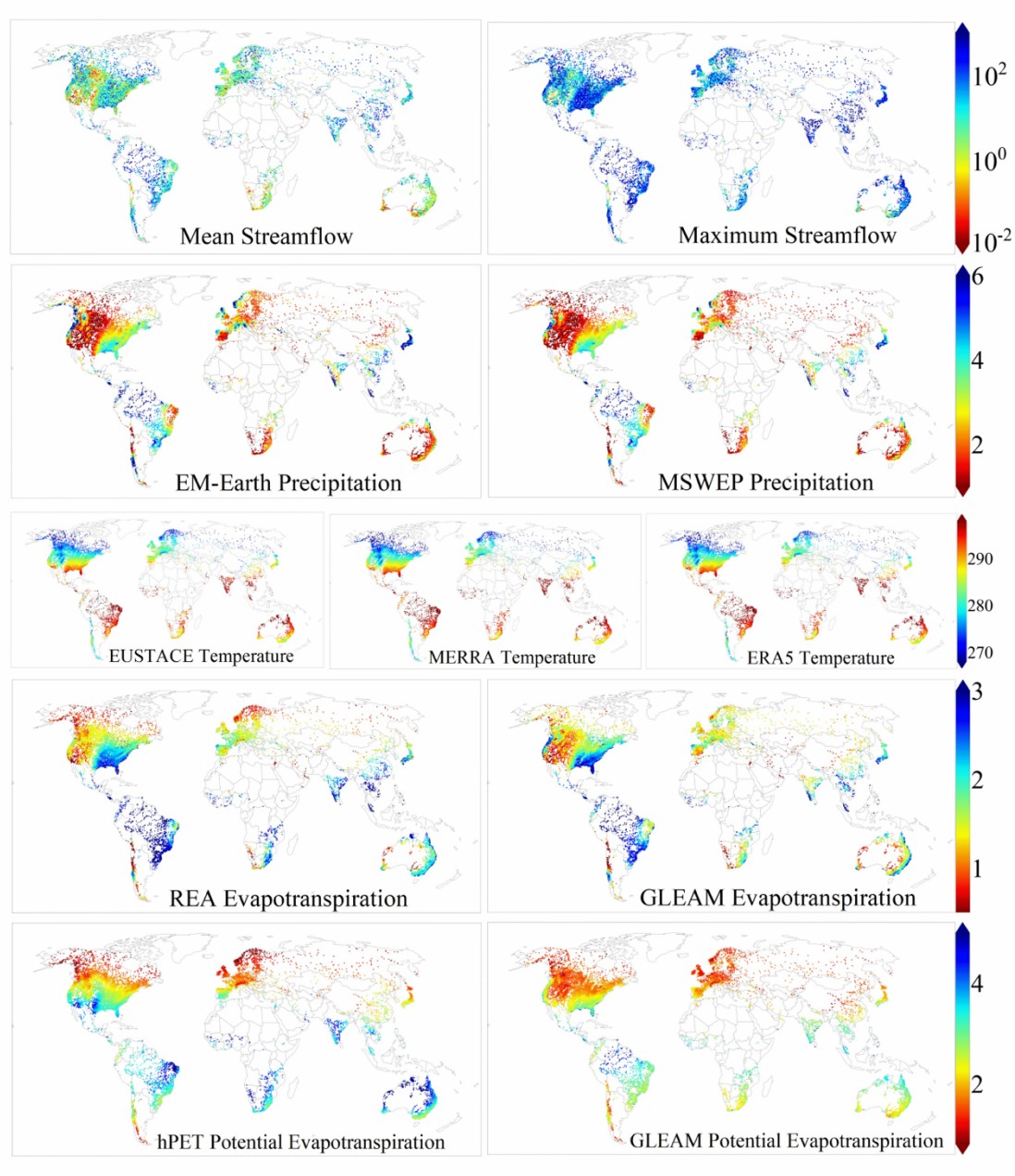


**Figure A1** Spatial distribution of streamflow indices (row 1, m³/s), precipitation (row 2, mm/day), 2 m
air temperature (row 3, K), actual ET (row 4, mm/day), potential ET (row 5, mm/day).

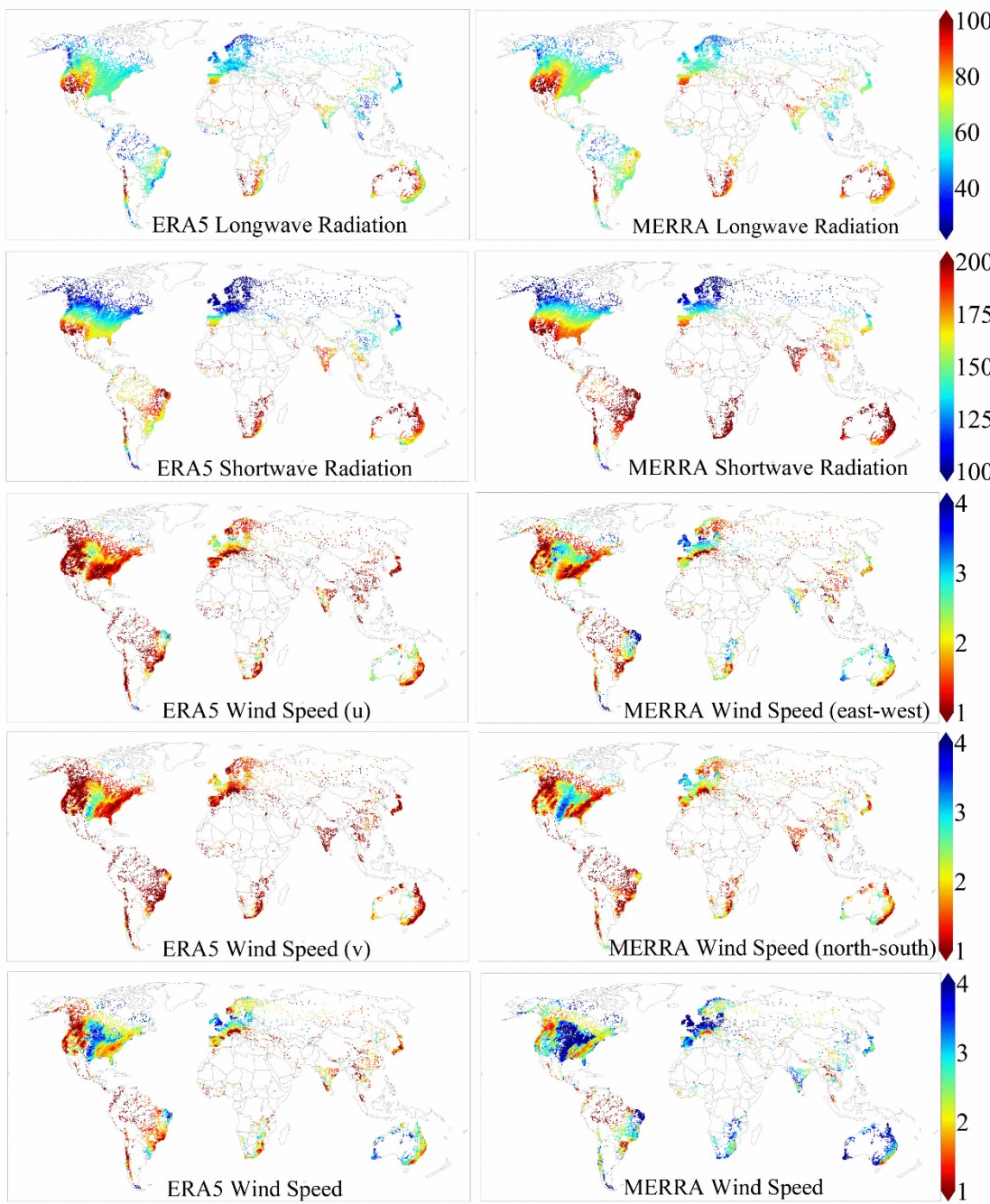

**Figure A 2** Spatial distribution of longwave radiation (row 1, W/m$^2$), shortwave radiation (row 2, W/m$^2$), wind u- (row 3, m/s) and v- components (row 4, m/s) and the wind speed (row 5, m/s).

## B. Validation results of watershed areas

The validation results with BOM, HYDAT, and GRDC on watershed areas are plotted in **Figure B1** and **B2**, where the mismatches between GSHA areas and the officially reported areas are shown. Before removing the mismatched watersheds, their correlation coefficients are 0.960, 0.840, 0.709, respectively, as showm in **Figure B1 (a), (b), (c).** After removing the mismatched watersheds, correlation coefficients for all three agencies reach 0.999, as shown in **Figure B1 (d), (e), (f)**. As we

traced the MERIT Basins (Lin et al., 2019) for our watershed delineation, the mismatches are
believed to occur when the gauge locates in the vicinity of the intersection point of a river reach and
its main stream, which makes it difficult to decide which reach the gauge belongs to while matching
the gauge to the MERIT river network. This explains why in **Figure B1** most of the mismatches
appear at relatively small areas. As we do not have access to all official watershed areas, and **Figure**
**B1 (a), (b), (c)** suggest that matching qualities differ among the agencies, to simply remove the
mismatched watersheds or to modify them might put the samples in the dataset under an unfair
standard. Additionally, some agencies such as GRDC experienced some updates of their gauige
locations and upstream areas, thus watershed boundaries in all datasets mentioned might come with
uncertainties. Therefore, we gave the watersheds as "unverified", "verified match", and "verified
mismatch" identifiers to allow users to flexibly filter the watersheds.

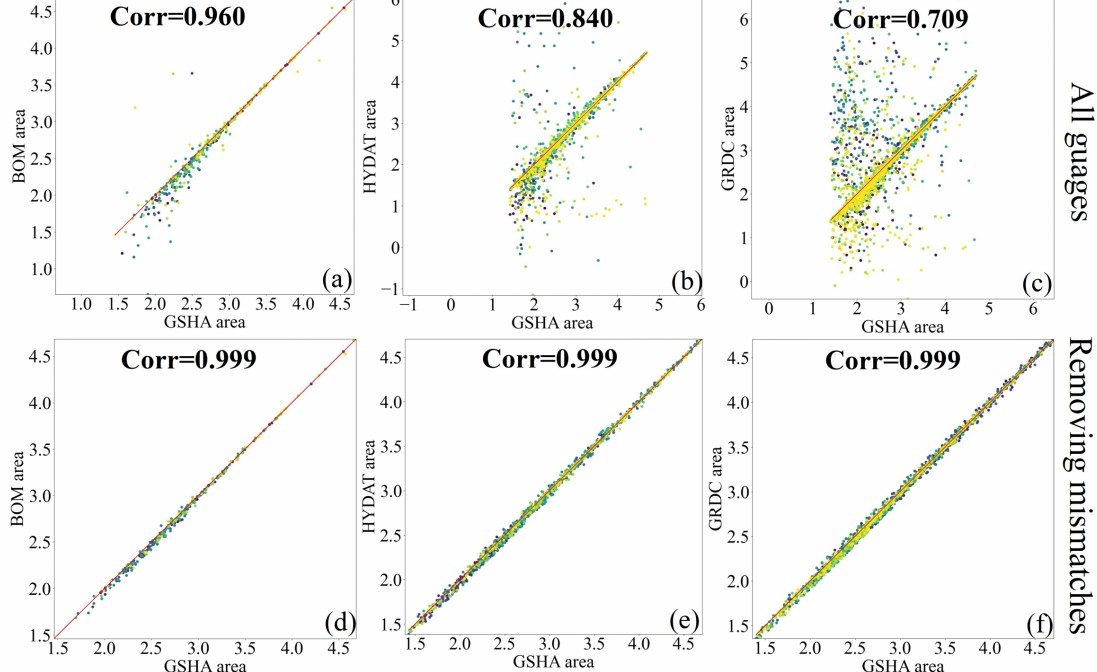

**Figure B1** Validation of GSHA with officially reported areas of BOM (a, d), HYDAT (b, e), and GRDC
(c, f). Subfigures (a) to (c) are the results before removing the mismatched watersheds, and subfigures
(d) to (f) represent results after removing the mismatched watersheds. The Pearson correlation coefficient
are represented by "Corr" in the figure. The areas are represented by the unit of (log10 km2).

## C. Potential evapotranspiration uncertainty

The spatial and numerical distributions of potential evapotranspiration (PET) uncertainties are
illustrated in **Figure C1** and **Figure C2**. PET uncertainty is high compared with other variables (see
5.2 section). The majority of high PET uncertainty watersheds are in dry areas, but since it is
calculated from meteorological variables, exceptions exist for palces including eastern Pacific coast,
where the climate is dry but PET uncertainty is low, and India, which is located in a wet climate
zone but has high PET uncertainty. As demonstrated by **Figure C3**, PET uncertainty do not decrease
with the increase of watershed area, probably because PET is calculated from various variables, and
the calculation over large watersheds involves more uncertainties for individual grids.

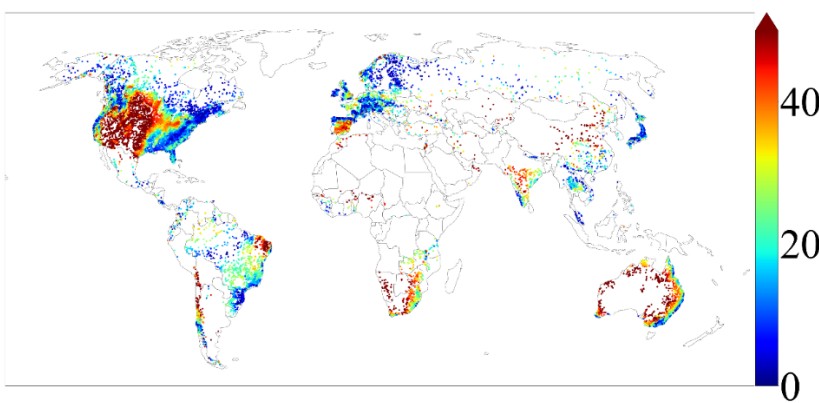

**Figure C1** Spatial pattern of potential evapotranspiration (PET) uncertainty.

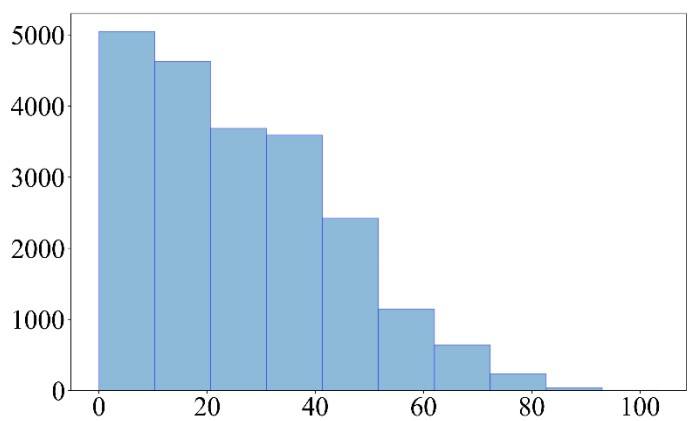

**Figure C2** Numerical distribution of PET uncertainty.

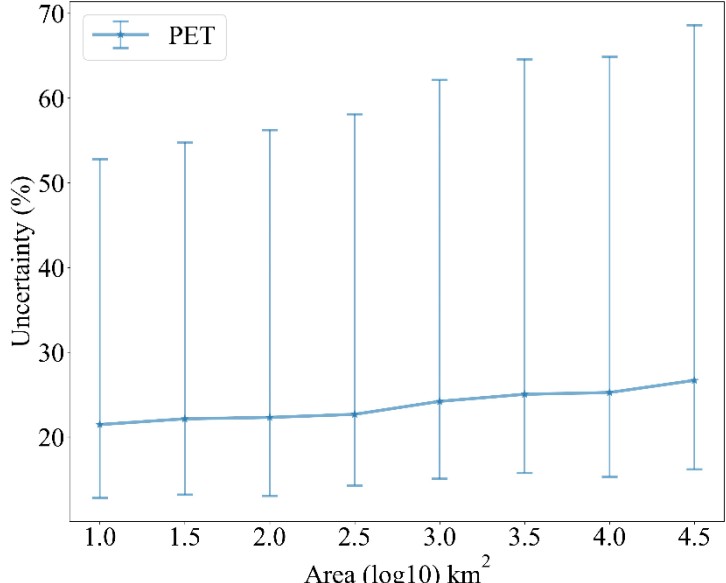

**Figure C3** Relationship of PET uncertainty to watershed area.

# Author contribution

Conceptualization: PL. Investigation: ZY, PL, RR, GA, XL. Data curation: ZY, RR, XL, PL, ZZ, SC. Funding acquisition: PL. Writing - initial: ZY, PL. Writing - Review and Editing: PL, ZY, GA, RR, XL.

# Data and Code Availability

GSHA v1.0 is openly available at https://doi.org/10.5281/zenodo.8090704 and https://doi.org/10.5281/zenodo.10127757. The codes involved in the workflow to generating GSHA will be available upon reasonable requests to the corresponding author.

# Competing interests

The authors declare no conflict of interest.

# Acknowledgements

This study is supported by the National Key Research and Development Program (2022YFF0801303), the Yunnan Provincial Basic Research Project-Science and Technology Special Project of Southwest United Graduate School (#305107035054), the Natural Science Foundation of China (42371481, 42175178), and the Fundamental Research Funds for the Central Universities to Peking University (#7100604136).

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
