# Peer review of "A Synthesis of Global Streamflow characteristics, Hydrometeorology, and"

_Earth System Science Data, 2023_

## Referee Comment (RC2)

**General Comment**

Yin et al generated a comprehensive hydrometeorological dataset at global scales. Compared to existing dataset, GSHA includes more variables and corresponding uncertainty estimates. It represents a significant contribution for large sample hydrology datasets and is very useful for data-driven hydrological applications, and calibration/validation of large-scale hydrological models and Earth system models. It is suitable to be published at Earth System Science Data. However, I recommend a major revision is needed before it is ready to be published. Please find my detailed comments in the following.

**Major Comments #1**

Line 163 – Line 170: The authors should clarify how they aggregated the daily streamflow to annual streamflow. There could be missing data for the streamflow in any year, and the missing days and the number of the missing days are not the same in different years. How the authors addressed the data gaps in the daily streamflow? For example, is there a criterion for the number of available days in a year that used to filter "good" years?

In addition, I think monthly streamflow indices are more useful for modelers to calibrate and validate models. For example, previous studies have used the monthly time series from GSIM to calibrate large scale hydrological models and Earth system models.

**Major Comments #2**

There is a lack of validation of watershed delineation. The watershed delineation could be one of the most important characteristics of GSHA, as many other variables were extracted based on watershed boundary. I think the flow directions may be carefully validated in previous study, but it is important to validate the delineated watershed boundary. For example, most gauges reported watershed boundary or drainage area, which can be used as benchmark.

**Major Comments #3**

One of the novelties of GSHA comparing to existing large sample hydrology datasets is GSHA provide the uncertainty analysis for the selected variables. But I think current description of uncertainty estimate is not clear, and the method is not comprehensive. Specifically,

Line 354-Line358: I don't think Eq (1) represents the uncertainty of the meteorological variables. As $X_{max}$ and $X_{min}$ represent the maximum and minimum values of the extracted variables from each individual dataset, $X_{max} - X_{min}$ is more linked to the natural variability instead of uncertainty of that dataset. For the example of temperature, if we have a dataset give us $X_{max} = 35°C$, $X_{min} = -5°C$, and $\bar{X} = 10°C$. Is the uncertainty of this dataset being $\frac{35-(-5)}{10} \times 100\% = 400\%$? And please further explain why the range of Eq (1) is between 0 and 200%.

However, based on the results in Figure 6, I think $X_{max}$ and $X_{min}$ are derived from all the datasets? I think the authors should further clarify the definition of uncertainty. In addition, $X_{max} - X_{min}$ cannot capture the uncertainty in the temporal variability. It is possible for two

datasets capture exact the same $X_{max}$ and $X_{min}$, but have different distribution. Thus, $X_{max} - X_{min}$ is not a good metric for analyzing the uncertainties from different datasets. I suggest the authors to include more metrics in the uncertainty analysis.

**Specific Comments**

Line 160: upstream drainage  area

The section numbers from 3.2 to 3.7 were wrong.

Line 212: Need to define "shorter record length" explicitly.

Line 216: CHP is defined in Table 3. But I think it is better to give the full name in the main text as well.

Line 318-Line 325: How you match the dams in GeoDAR to GSHA? Is it possible for a watershed to have several dams? How about the watershed that doesn't have a dam from GeoDAR?

Table 4: I think MSWEP is at spatial resolution of $0.1° \times 0.1°$. Please double check.

Figure 5a, b, and c: Are the X and Y axis normalized? I suggest the authors to plot the original data (e.g., in $m^3/s$) to demonstrate that no system errors were introduced during the processing of GSHA. In addition, the authors should explain why the comparison of some watersheds are very off from the 1:1 line. In my understanding, both GSIM and GSHA were derived from gauge observations for the streamflow indices. Therefore, same gauge measurements should be used at the same watershed in both GSIM and GSHA. Is the significant difference caused by (1) different gauges were used, or (2) different method was applied to address the data gaps in the gauge measurements (see my Major Comments #1), etc. Overall, I think it is useful for the authors to further explain the significant discrepancies in those gauges.

Figure 7: I don't think the decline of uncertainty as the watershed area increases is obvious for longwave radiation.

Line 517 – Line 529: The analysis of runoff coefficient and its changing trend in the past few decades is very interesting and is very critical for us to understand response of hydrological cycle to global warming. Such analysis with observed streamflow is more convincing than model simulations, which can be highly biased. I believe there exist some other studies focusing on this topic, such as runoff trend in this historical period. I wonder if the authors can give more discussion for this analysis and include more references.

---

## Community Comment (CC2)

Dear Ather,

Thanks for your comment. We checked the links you mentioned. The Chinese National Real-time and Water Situation Database website can be fully accessed through Internet in China. The choice of web browsers or the restricted access for researchers outside of China may be the possible reasons for not able to opening it. Figure 1 provides a screenshot of the website.

For the Japanese link, it only brings you to parts of the dataset through direct links. Figure 2 to Figure 7 show the processes on how to access daily streamflow data for one Japanese gauge for one year, following the link we provided in the manuscript. Therefore, the general link http://www1.river.go.jp/ could lead users to the entire dataset. For the Spanish dataset, the original link is indeed not straightforward as it directs users to all the yearbooks of Spain Annuario de Aforos. After checking, we decided that https://ceh.cedex.es/anuarioaforos/demarcaciones.asp is a better link to the gauge information, which will be modified in the revised version of the manuscript.

Thanks again for your comment.

| Tuesday, August 2023, 8 | | | | | Welcome to the National Water and Rain Information Website! | | |
|---|---|---|---|---|---|---|---|

全国水雨情信息

| Home | Flood and drought warning | Water inquiries |
|---|---|---|
| | Water conservancy scenic area | |

| | | | Focus on rain | Daily rainfall |
|---|---|---|---|---|
| Map query | Big rivers | Large reservoirs | conditions | nationwide |

Real-time water conditions in major rivers across the country

Prepared by: Information Center of the Ministry of Water Resources          Statement date: 2023-08-01

| drainage basin | District | River name | Station name | Time | Water level (m) | Flow rate ($m^3$/sec) | Alert water level (m) |
|---|---|---|---|---|---|---|---|
| Songhua River | Jilin Province | Nenjiang | Dazhao | 08-01 14:00 | 129.44 | -- | 131.74 |
| Heilongjiang | Jilin Province | Songhua River | Buyeo | 08-01 14:00 | 130.86 | 997 | 133.56 |
| Inland rivers and lakes | Xinjiang Uygur Autonomous Region | Bortala River | hot spring | 08-01 14:00 | 1320.86 | 23 | -- |
| Inland rivers and lakes | Xinjiang Uygur Autonomous Region | Jinghe | Seihe Pass (3) | 08-01 14:00 | 614.33 | 31 | -- |
| Manas River Lake | Xinjiang Uygur Autonomous Region | Manas river | Kenswat | 08-01 14:00 | 855.32 | 188 | -- |
| Inland rivers and lakes | Xinjiang Uygur Autonomous Region | Urumqi River | Heroes Bridge (II) Hydrological Station | 08-01 08:00 | 1756.18 | 22 | -- |
| Inland rivers and lakes | Xinjiang Uygur Autonomous Region | Reclamation of the river | Reclamation of the River (3) | 08-01 14:00 | 1504.93 | 1 | -- |
| Inland rivers and lakes | Xinjiang Uygur Autonomous Region | Alla Gou | Alagou (II) | 08-01 14:00 | 784.14 | -- | -- |
| The rivers of Hami and Turpan regions | Xinjiang Uygur Autonomous Region | Head ditch | Head ditch | 08-01 14:00 | 1423.33 | -- | -- |

Figure 1 A screenshot of the Chinese National Real-time and Water Situation Database.

Figure 2 A screenshot on how to search for data for a Japanese gauge.

Figure 3 A screenshot on how to select a Japanese gauge.

Figure 4 A screenshot on how to access the details of a Japanese gauge.

**Information of the Hydrological and Water Quality Institute**

| | |
|---|---|
| The name of the test | Hanaishi(はないし) |
| Speculative items | Water level flow |
| Measure the mark | 301051281102020 |
| Name of the water system | Go-Shiri Besukawa |
| River name | Go-Shiri Besukawa |
| location | 226 in present-day Kanamachi Nakazato, Setana District, Hokkaido |
| Latitude degree | Latitude 42 degrees 25 minutes 42 seconds East 140 degrees 09 minutes 13 seconds |
| 最新の零點高 | T.P. 0.000m |
| 水位計読み値の変更履歴※ | -0.070m | 2005/01/01 0:00 ~ |

※水準点の標高が見直され、水位観測所の水平基標が修正されたことに伴い、
実際の水位は変わらないのですが水位計読み値の変更を行いました。

Figure 5 A screenshot on how to search for the hydrological variable-of-interest selected Japanese gauge.

Figure 6 Decide the temporal extent.

Figure 7 Access the flow data.

---

## Author Response (AR1)

**We thank the reviewers and the editor for handling our paper and offering constructive comments. Below please find our point-by-point response.**

**Reviewer #1**

**Comment 1:** It would be very helpful to include stream order, COMID and NextDownID based on the MERIT hydrography. The thinking here is that many catchments are nested because the dataset is gauge-based. Including the suggested columns/fields could enable the users to incorporate upstream-downstream relationship in their analyses, for instance, where scaling behavior is key.

Thank you for your suggestion. We added another file "GSHA_MERITinfo.csv" in the "/Global_files/" directory with COMID, order, and NextDownID fields from the MERIT database. We also included the uparea field from MERIT as a comparison of our watershed boundaries.

**Comment 2:** Minor comment: the writing mixed present and past tenses, causing confusion on which actions are taken by the authors and which are taken in previous studies. For instance, the sentence from L288 to L292 is grammatically ill and hard to understand.

Thanks for your comment. We checked the verbs in the article and modified the tense of the wrong sentences. The description in 3.4.1 Meteorology datasets section involves information of many dataset names and might be confusing. Therefore, we modified the entire section to shorter sentences to clarify the descriptions. We also modified some other grammar mistakes in the text. We are sorry for the inconvenience.

**Comment 3:** The dataset involves merging several datasets – taking weighted average. I expect a brief introduction of the motivation/justification of applying certain method, for instance, Lu et al.,2021. This is important because it actually matters whether to merge several data sources or to simply provide their individual values. To me, merging is only meaningful when estimation error from individual merged component is informed. This is why the authors need to better justify/explain the merging of multiple datasets.

Thank you for your comment. We did not use the weighted average to merge the datasets, which is different from the purpose of Lu et al., 2021. The area-weighted approach was used to extract grid data to watersheds. Because the grids are fully or partially contained by the watershed boundaries, it is not accurate to simply calculate the arithmetic mean of all the grids intersecting with the watershed boundary. Therefore, we calculate a weight for each grid representing the proportion of grid contained by the watershed boundary. This follows the method adopted by Addor et al., 2017, which is the foundation of CAMELS datasets. For dataset merging, we agree that simply merging the datasets by their mean values without further estimating errors is irresponsible. Therefore, we treat each dataset as an independent estimation of the

variable and did not merge them for an averaged value. We added "based on the proportion of the grid area contained in the basin boundary" in line 348 to clarify the spatial aspect of this issue.
* * *
**Reviewer #2**

**Major Comments #1**

Line 163 – Line 170: The authors should clarify how they aggregated the daily streamflow to annual streamflow. There could be missing data for the streamflow in any year, and the missing days and the number of the missing days are not the same in different years. How the authors addressed the data gaps in the daily streamflow? For example, is there a criterion for the number of available days in a year that used to filter "good" years?

Thank you for your comment. We aggregated the daily streamflow by calculating their annual indices, such as annual mean, maximum, percentiles, as well as temporal characteristics such as maximum flood occurrence date, duration of high and low flow events. Therefore, we describe our data as streamflow characteristics instead of annual streamflow. **The dataset includes a "valid observation days" field**, which describes the number of days with available daily streamflow in the corresponding year, **as well as a "Q=0 days" field** representing the number of days with runoff measurement equal to 0. The data were not filtered or selected based on any criterion set by the authors, because **we would like to let the users decide how many available or non-zero measurements define a "good" year to them based on their research purposes and scales.** To make this clearer, we modified the sentence in lines 170-171 to "We also include numbers of zero observations and valid samples to allow flexible data screening by the users."

In addition, I think monthly streamflow indices are more useful for modelers to calibrate and validate models. For example, previous studies have used the monthly time series from GSIM to calibrate large scale hydrological models and Earth system models.

Thanks for the comment. We now publicize the monthly indices of gauge observations (except for some transboundary watersheds) by calculating the monthly mean, maximum, percentiles, max flow occurrence date, number of days with Q=0, and valid observation days of the watersheds after 1979, and **attached the files can be found at https://zenodo.org/records/10127757**.

**Major Comments #2**

There is a lack of validation of watershed delineation. The watershed delineation could be one of the most important characteristics of GSHA, as many other variables were

extracted based on watershed boundary. I think the flow directions may be carefully validated in previous study, but it is important to validate the delineated watershed boundary. For example, most gauges reported watershed boundary or drainage area, which can be used as benchmark.

Thanks for your comment. Since we found out we do not have access to officially reported areas of all watersheds from agency websites, we validated our watershed areas for Australian Bureau of Meteorology 2022 (BOM), Canada National Water Data Archive 2022 (HYDAT), and The Global Runoff Data Centre 2022 (GRDC). The validation results are plotted in **Figure R1**.

There are indeed mismatches between GSHA areas and the officially reported areas by the agencies. As we used the MERIT Basins (Lin et al., 2019) for watershed dissolving, we do not question the sub-watersheds used in our delineation. After we compared our watershed area with officially reported area, it is found that some mismatches might occur when the gauge appears in the vicinity of the intersection point of a river reach and its main stream, which makes it difficult to decide which reach the gauge belongs to while matching the gauge to the MERIT river network. This explains why in Figure 1 most of the mismatches appear at relatively small areas.

To address this issue, we set the criteria of mismatched watershed as: (1) the area difference being over 20% of the officially reported area, and (2) the area ratio being over 10 times. **Under this criterion, 1.9% of BOM watersheds, 4.7% of HYDAT watersheds and 8.9% of GRDC watersheds are mismatched**, as plotted in **Figure 1** (a) to (c). After removing these watersheds, (d) to (f) show very good match of the watershed areas, with correlation coefficients reaching 0.999, suggesting that the remaining watersheds match with the officially recorded areas well.

Therefore, we decide to make the following modifications:

(1) **Add the area validation** to 3.7 Validation, 4.1 Technical Validation and Appendix B sections to inform the readers of the issue and the casual factors;
(2) **Add a flag field in the watershed list** of the dataset to describe the watersheds as "unverified", "verified match", and "verified mismatch". As we do not have access to all official watershed areas, to simply remove the mismatched watersheds or to modify them might put the samples in the dataset under an unfair standard.
(3) We do not identify our mismatched gauges as "wrong" because whether our delineations are incorrect remains to be investigated, and after further check with other sources we will update our conclusions in the next version of GSHA. At the end of October, approximately 400 GRDC gauges updated their coordinates and some of them have experienced major deviations. After inquiring GRDC on this issue, we received their response as "*We occasionally receive updates on the metadata from the National Hydrological Services (NHS). This explains the smaller deviations, as it includes updates of longitude and latitude, as well as altitude and the size of the catchment area. The larger deviations occurred for the following reason. While*

*recalculating the station-based catchment areas, there were some stations for which an exact derivation was not possible. The reason for this was that the coordinates were incorrect or inaccurate*". This suggests that data from some agencies come with errors and uncertainties, and we will follow up on these updates to obtain more accurate information.

[Figure]

Figure R1 Validation of GSHA with officially reported areas of BOM (a, d), HYDAT (b, e), and GRDC (c, f). Subfigures (a) to (c) are the results before removing the mismatched watersheds, and subfigures (d) to (f) represent results after removing the mismatched watersheds. The Pearson correlation coefficient are represented by "Corr" in the figure. The areas are represented by the unit of (log10 km²).

**Major Comments #3**

One of the novelties of GSHA comparing to existing large sample hydrology datasets is GSHA provide the uncertainty analysis for the selected variables. But I think current description of uncertainty estimate is not clear, and the method is not comprehensive. Specifically, Line 354-Line358: I don't think Eq (1) represents the uncertainty of the meteorological variables. As $X_{max}$ and $X_{min}$ represent the maximum and minimum values of the extracted variables from each individual dataset, $X_{max}$ and $X_{min}$ is more linked to the natural variability instead of uncertainty of that dataset. For the example of temperature, if we have a dataset give us $X_{max}$= 35°C, $X_{min}$ = -5°C, and X = 10°C. Is the uncertainty of this dataset being 400%? And please further explain why the range of Eq (1) is between 0 and 200%.

However, based on the results in Figure 6, I think $X_{max}$ and $X_{min}$ are derived from all the datasets? I think the authors should further clarify the definition of uncertainty. In addition, $X_{max}$ and $X_{min}$ cannot capture the uncertainty in the temporal variability. It is possible for two datasets capture exact the same $X_{max}$ and $X_{min}$, but have different distribution. Thus, $X_{max}$ and $X_{min}$ is not a good metric for analyzing the uncertainties from different datasets. I suggest the authors to include more metrics in the uncertainty analysis.

Thanks for the comment. The uncertainty we calculate represents the discrepancy between long-term means of the datasets, instead of the differences of each value in the time series. The $X_{max}$ and $X_{min}$ values are the maximum and minimum values in the **dataset ensembles (in our dataset two to three members included), rather than the max and min values in the temporal series**. We use this estimate to represent uncertainty of the mean value. Therefore, the distributions and variances inside each dataset are not considered. We understand that uncertainty should be represented by a range around the true value of the variable, but we do not know the true values of each variable at each particular date, and daily estimates from the datasets can be very biased. Therefore, we believe uncertainty range represented by discrepancy of the long term mean can be more meaningful compared to a time series of daily differences. 200% uncertainty occurs when one dataset $X_{min} = 0$ and $X_{max} > 0$. As we use K as temperature unit, there will be no negative value in the data.

To clarify this concept, we modified the sentence in line 368-371 as "We also provide uncertainty estimates of the meteorological variables by calculating the long-term mean of each dataset in each watershed, where the discrepancy between the maximum and minimum among the data sources ($X_{max}$ and $X_{min}$) as a percentage of their mean ($\bar{X}$) was used in the uncertainty estimation".

**Specific Comments**

Line 160: upstream drainage  area

Thank you for this comment. We modified the mistake and checked all descriptions of watershed area.

The section numbers from 3.2 to 3.7 were wrong.

We changed the wrong section numbers to 3.3 to 3.8.

Line 212: Need to define "shorter record length" explicitly.

We changed "shorter record length" to "fewer years of measurement" in line 215.

Line 216: CHP is defined in Table 3. But I think it is better to give the full name in the main text as well.

We added the full name of CHP at its first appearance in Line 219.

Line 318-Line 325: How you match the dams in GeoDAR to GSHA? Is it possible for a watershed to have several dams? How about the watershed that doesn't have a dam from GeoDAR?

We used the reservoir polygons in GeoDAR instead of the dam locations. To clarify the extraction process, we added the sentence "For reservoirs, we used the reservoir polygons in GeoDAR, which are spatially joined to GSHA watershed polygons. All the intersected reservoirs were considered contributory to the management of the corresponding watershed, and were used to calculate the total reservoir storage capacity and degree of regulation" in lines 351-354. For watersheds with multiple reservoirs, the sum of the capacities of the reservoirs were calculated. For watersheds with no reservoir, the capacity and DOR fields were set as empty. We manually checked a portion of the spatial join, and found the automatic spatial join approach to be reasonable.

Table 4: I think MSWEP is at spatial resolution of 0.1° × 0.1°. Please double check.

The version 2 of MSWEP is 0.1° × 0.1° resolution. However, our attempt of this research started before 2019. Therefore, MSWEP v1 with a 0.25° × 0.25° resolution were used. We will extract MSWEP v2 values in the updated version of GSHA in the near future.

Figure 5a, b, and c: Are the X and Y axis normalized? I suggest the authors to plot the original data (e.g., in m3/s) to demonstrate that no system errors were introduced during the processing of GSHA. In addition, the authors should explain why the comparison of some watersheds are very off from the 1:1 line. In my understanding, both GSIM and GSHA were derived from gauge observations for the streamflow indices. Therefore, same gauge measurements should be used at the same watershed in both GSIM and GSHA. Is the significant difference caused by (1) different gauges were used, or (2) different method was applied to address the data gaps in the gauge measurements (see my **Major Comments #1**), etc. Overall, I think it is useful for the authors to further explain the significant discrepancies in those gauges.

Thanks for this comment. The X and Y axes are log10 results of the original data, since the original data plot can be dominated by a few very large observations, as shown in **Figure R2**. Therefore, in order to clearly show the distribution of the majority of the samples, we used the log10 of original data. We are sorry for not clarifying this in the text. We added "The unit of X and Y axes in (a), (b). and (c) is long10 m3/s" in the caption of Figure 5.09.

We matched the gauges by their latitudes and longitudes, each point should represent the pair of the same gauge. However, the location matching might confuse a small proportion of very close gauges. Therefore, it is possible that the different gauges used cause deviations of validation results, and **we think locational error is the most**

**significant factor causing the problem**. However, currently we do not have a proper method to find out which gauge pairs are wrong based on ids and locations, thus we plotted all pairs in the validation figures. For data selection, GSIM suggested that "Given that data quality requirements can vary substantially, it will remain the work of individual users to establish selection criteria for each study, thereby finding a trade-off between data quantity (number of gauges) and data quality (record length, missing periods)" (Gudmundsson et al., 2018), which is consistent with our decision not to filter the observations as mentioned in the reply of Major Comments #1. However, according to the time step in the GSIM file, the first time step and last time step are usually 31$^{st}$ Dec., apart from some missing values, while we did not process our data that way. This might cause some discrepancies, but with monthly indices provided, we believe more accurate analysis can be carried out. We added these two reasons in the 4.1 Technical Validation section to inform the readers of these causes of differences.

[Figure]

Figure R2 Validation of GSHA with GSIM streamflow 90 percentile. The red line is the 1:1 line, while the orange dotted line is the fitting line of the scatter points.

Figure 7: I don't think the decline of uncertainty as the watershed area increases is obvious for longwave radiation.

We agree with this comment and modified the description as "The most obvious decline comes from ET (green), which is highly dependent on the land surface conditions and are significantly affected by land surface spatial heterogeneity, thus benefiting the most from spatial averaging for large river basins. Longwave radiation uncertainty (red) experiences a moderate decline, mainly due to its connection with land surface complexity and cloud conditions" in lines 483-486.

Line 517 – Line 529: The analysis of runoff coefficient and its changing trend in the past few decades is very interesting and is very critical for us to understand response of

hydrological cycle to global warming. Such analysis with observed streamflow is more convincing than model simulations, which can be highly biased. I believe there exist some other studies focusing on this topic, such as runoff trend in this historical period. I wonder if the authors can give more discussion for this analysis and include more references.

Thanks for the comment. We added some discussions on runoff coefficient (RC) analysis considering land cover change in section 4.4, which are largely regional studies or focusing on individual cases. Additionally, our investigation suggested that such analysis incorporating water consumption and other human modifications, especially on large scale, are still insufficient. Therefore, we believe there is still a gap on the identification of large-scale patterns of RC trend and its attribution. We will follow up on this topic and try to identify signals of water availability change and the casual factors based on observations.

---

## Referee Report (RR1)

General Comment:

Thank you for your comment. We aggregated the daily streamflow by calculating their annual indices, such as annual mean, maximum, percentiles, as well as temporal characteristics such as maximum flood occurrence date, duration of high and low flow events. Therefore, we describe our data as streamflow characteristics instead of annual streamflow. **The dataset includes a "valid observation days" field**, which describes the number of days with available daily streamflow in the corresponding year, **as well as a "Q=0 days" field** representing the number of days with runoff measurement equal to 0. The data were not filtered or selected based on any criterion set by the authors, because **we would like to let the users decide how many available or non-zero measurements define a "good" year to them based on their research purposes and scales.** To make this clearer, we modified the sentence in lines 170-171 to "We also include numbers of zero observations and valid samples to allow flexible data screening by the users."

**Comments:** Thank you for addressing my comments. However, I think the annual indices with number of available daily/sub-daily data should be reported as 'NaN' or removed. If we have both Year A and Year B has the same number of daily data, e.g., ~180 days, to derive the annual indices. But Year A is available from Jan to Jun, while Year B is available from Jul to Dec. Such inconsistency will result in bias for calibrating/validating model simulation because they represent streamflow characteristic from different seasons. Therefore, although the users can decide if a year is "good" or not, they will not know if all the good years are consistent in the time period.

Thanks for your comment. Since we found out we do not have access to officially reported areas of all watersheds from agency websites, we validated our watershed areas for Australian Bureau of Meteorology 2022 (BOM), Canada National Water Data Archive 2022 (HYDAT), and The Global Runoff Data Centre 2022 (GRDC). The validation results are plotted in **Figure R1**.

**Comments:** I appreciate the authors' efforts to validate the watershed delineation based on my comment. I believe the contributing area should be reported for each gauge, at least from USGS. For example, the author can find the drainage area at this USGS gauge: https://waterdata.usgs.gov/monitoring-location/07374000/#parameterCode=00065&period=P7D&showMedian=true. I understand that GRDC gauge coordinates may be highly uncertain, so USGS gauges could be a good benchmark, which is at higher quality.

Thanks for the comment. The uncertainty we calculate represents the discrepancy between long-term means of the datasets, instead of the differences of each value in the time series. The

$X_{!"\#}$ and $X_{!\$\%}$ values are the maximum and minimum values in the **dataset ensembles (in our dataset two to three members included), rather than the max and min values in the temporal series**. We use this estimate to represent uncertainty of the mean value. Therefore, the distributions and variances inside each dataset are not considered. We understand that uncertainty should be represented by a range around the true value of the variable, but we do not know the true values of each variable at each particular date, and daily estimates from the datasets can be very biased. Therefore, we believe uncertainty range represented by discrepancy of the long term mean can be more meaningful compared to a time series of daily differences. 200% uncertainty occurs when one dataset $X_{min} = 0$ and $X_{max} > 0$. As we use K as temperature unit, there will be no negative value in the data.

**Comments**: Do you mean for the case $X_{min} = 0$ and $X_{max} > 0$, the $\bar{X} = \frac{X_{max}}{2}$? Do you assume the variable $X$ varies linearly from $X_{min}$ to $X_{max}$?

We matched the gauges by their latitudes and longitudes, each point should represent the pair of the same gauge. However, the location matching might confuse a small proportion of very close gauges. Therefore, it is possible that the different gauges used cause deviations of validation results, and **we think locational error is the most significant factor causing the problem**. However, currently we do not have a proper method to find out which gauge pairs are wrong based on ids and locations, thus we plotted all pairs in the validation figures. For data selection, GSIM suggested that "Given that data quality requirements can vary substantially, it will remain the work of individual users to establish selection criteria for each study, thereby finding a trade-off between data quantity (number of gauges) and data quality (record length, missing periods)" (Gudmundsson et al., 2018), which is consistent with our decision not to filter the observations as mentioned in the reply of Major Comments #1. However, according to the time step in the GSIM file, the first time step and last time step are usually 31st Dec., apart from some missing values, while we did not process our data that way. This might cause some discrepancies, but with monthly indices provided, we believe more accurate analysis can be carried out. We added these two reasons in the 4.1 Technical Validation section to inform the readers of these causes of differences.

**Comments**: Except the location and id, the contributing area can be used as the third criterion for paring the gauge in both GSIM and GSHA. Specifically, if the contributing area are not the same, there is a high probability that not the same gauge is used in GSIM and GSHA for comparison.

In addition, I don't understand how the time step impact the annual streamflow indices, e.g., p90 that is reported in Figure R2. Should the estimate of the annual streamflow indices have based on 365 or 366 daily streamflow (if the data is available for the whole year)?

---

## Author Response (AR2)

**Comment:** However, I think the annual indices with number of available daily/sub-daily data should be reported as 'NaN' or removed. If we have both Year A and Year B has the same number of daily data, e.g., ~180 days, to derive the annual indices. But Year A is available from Jan to Jun, while Year B is available from Jul to Dec. Such inconsistency will result in bias for calibrating/validating model simulation because they represent streamflow characteristic from different seasons. Therefore, although the users can decide if a year is "good" or not, they will not know if all the good years are consistent in the time period.

Thank you for this comment. We acknowledge that the temporal distribution of available data throughout the year can influence analysis and interpretation, thus **we added a new field "month with nan>10 days" in the yearly indices table, which includes the list of the months with over 10 days of NaN measurement (see examples from the last columns of Tables R1 and R2).** We did not set the annual indices as "NaN" directly for years with a fixed number of available daily observations because the purposes of data users can be very different, and we would like to make the selection criteria as flexible as possible to allow for more potential utilizations. For instance, Table R1 shows a gauge where AMF usually occurs in summer. Therefore, despite that data for 2001-2005 are not intact, the AMF indices are generally reliable, while mean, median and low flow data are not usable. For another case in Table R2, although observation days exceed 250 in 1988 and 2021, streamflow data for consecutive three months are missing, thus annual means of these two years are still not equally representative as other years. We hope the additional field of the data table satisfies the quality control while leaving flexibility. We've revised our dataset in the uploaded file.

| Year | mean | maximum (AMF) | AMF occurrence date | number of days with Q=0 (days) | valid observation days (days) | month with nan>10 days |
|------|------|---------------|---------------------|-------------------------------|-------------------------------|------------------------|
| 2001 | 49.48039 | 691 | ['2001/07/29'] | 0 | 181 | [1, 2, 3, 4, 11, 12] |
| 2002 | 49.65326 | 795 | ['2002/06/28'] | 0 | 184 | [1, 2, 3, 4, 11, 12] |
| 2003 | 153.8382 | 1115 | ['2003/08/16'] | 0 | 185 | [1, 2, 3, 4, 11, 12] |
| 2004 | 98.21307 | 1579.714 | ['2004/07/17'] | 0 | 185 | [1, 2, 3, 4, 11, 12] |
| 2005 | 163.0907 | 940.25 | ['2005/09/28'] | 0 | 245 | [1, 2, 3, 4] |
| 2006 | 38.06584 | 715.25 | ['2006/07/05'] | 0 | 364 | [] |
| 2007 | 50.60067 | 820 | ['2007/07/14'] | 0 | 365 | [] |
| 2008 | 32.33534 | 468.4 | ['2008/07/24'] | 0 | 366 | [] |
| 2009 | 83.40167 | 909.3 | ['2009/06/19'] | 0 | 365 | [] |
| 2010 | 192.2253 | 4474.231 | ['2010/07/25'] | 0 | 365 | [] |
| 2011 | 40.71113 | 616.4815 | ['2011/09/15'] | 0 | 365 | [] |
| 2012 | 50.16656 | 335.5769 | ['2012/07/06'] | 0 | 366 | [] |
| 2013 | 23.58099 | 214.875 | ['2013/07/02'] | 0 | 365 | [] |
| 2014 | 12.67763 | 44.52917 | ['2014/10/02'] | 0 | 365 | [] |
| 2015 | 23.53541 | 123.16 | ['2015/05/04'] | 0 | 365 | [] |

Table R1 Selected from 62011800_China.csv.

| year | median | mean | maximum (AMF) | AMF occurrence date | number of days with Q=0 (days) | valid observation days (days) | month with nan>10 days |
|------|--------|------|---------------|---------------------|-------------------------------|-------------------------------|------------------------|
| 1985 | 0 | 1.727559 | 126.307 | 1985/11/3 | 307 | 365 | [] |

| | | | | | | | | |
|------|--------|----------|----------|------------|-----|-----|--------------|
| 1986 | 0 | 0.034907 | 6.177 | 1986/7/5 | 354 | 365 | [] |
| 1987 | 0 | 11.62648 | 775.377 | 1987/2/15 | 298 | 365 | [] |
| 1988 | 0 | 4.036762 | 230.29 | 1988/4/3 | 199 | 273 | [10, 11, 12] |
| 2005 | 0 | 0.038632 | 3.364 | 2005/6/23 | 200 | 212 | [1, 2, 3, 4, 5] |
| 2006 | 0 | 3.711151 | 488.019 | 2006/4/6 | 302 | 365 | [] |
| 2007 | 0 | 3.464066 | 338.672 | 2007/1/22 | 328 | 365 | [] |
| 2008 | 0 | 0.63915 | 81.378 | 2008/11/27 | 345 | 366 | [] |
| 2009 | 0 | 36.35345 | 1330.387 | 2009/1/9 | 277 | 365 | [] |
| 2010 | 0 | 6.590668 | 878.593 | 2010/1/8 | 285 | 365 | [] |
| 2011 | 0 | 4.40994 | 189.484 | 2011/1/18 | 250 | 365 | [] |
| 2012 | 0 | 0.789429 | 62.876 | 2012/3/16 | 328 | 366 | [] |
| 2013 | 0 | 0.138811 | 21.486 | 2013/11/29 | 345 | 365 | [] |
| 2014 | 0 | 2.281156 | 233.295 | 2014/3/1 | 311 | 365 | [] |
| 2015 | 0 | 2.899764 | 252.943 | 2015/12/28 | 335 | 365 | [] |
| 2016 | 0.0405 | 14.60942 | 589.812 | 2016/3/12 | 149 | 366 | [] |
| 2017 | 0 | 1.908384 | 145.384 | 2017/1/16 | 316 | 365 | [] |
| 2018 | 0 | 4.594542 | 673.417 | 2018/3/5 | 333 | 365 | [] |
| 2019 | 0 | 15.76645 | 1654.957 | 2019/3/28 | 268 | 365 | [] |
| 2020 | 0 | 2.696989 | 157.835 | 2020/1/27 | 315 | 366 | [] |
| 2021 | 0 | 1.02686 | 47.653 | 2021/2/13 | 211 | 250 | [10, 11, 12] |

Table R2 Selected from 001202A_BOM.csv.

**Comment:** I appreciate the authors' efforts to validate the watershed delineation based on my comment. I believe the contributing area should be reported for each gauge, at least from USGS. For example, the author can find the drainage area at this USGS gauge: hcps://waterdata.usgs.gov/monitoringloca Won/07374000/#parameterCode=00065&period=P7D&showMedian=true. I understand that GRDC gauge coordinates may be highly uncertain, so USGS gauges could be a good benchmark, which is at higher quality.

Thank you for your comment. We have actually validated the GSHA gauges against the HYDAT, GRDC, BOM, and USGS gauges and added the verification flag field in the dataset, but only the validation scatter plot of the previous three agencies were shown in the appendix. Here in Figure R1 we show the validation result of the USGS gauges. Correlation coefficient is 0.905 before removing the mismatched watersheds, and 0.999 after removing the mismatched watersheds. Based on the good match, we added the validation results of USGS areas in Figure B1 (Figure R2 in this reply file) in Appendix B in the latest revised version of the manuscript.

[Figure]

Figure R1 Validation of GSHA with officially reported areas of USGS gauges. Subfigure on the left is the result before removing the mismatched watersheds, and subfigure on the right is the result after removing the mismatched watersheds.

[Figure]

Figure R2 Validation of GSHA with officially reported areas of BOM (a, e), HYDAT (b, f), GRDC (c, g), and USGS (d, h). Subfigures (a) to (d) are the results before removing the mismatched watersheds, and subfigures (e) to (h) represent results after removing the mismatched watersheds. The Pearson correlation coefficient are represented by "Corr" in the figure. The areas are represented by the unit of (log10 km²).

**Comment:** Do you mean for the case $X$min = 0 and $X$max > 0, the $\bar{X} = \frac{X\text{max}}{2}$?

Yes. In our cases except for temperature, $X$min = 0 and $X$max > 0, thus $\bar{X} = \frac{X\text{max}}{2}$ and uncertainty equals to 200%. For temperature, both $X$min and $X$max > 0, and uncertainty is smaller than 200%.

Do you assume the variable $X$ varies linearly from $X$min to $X$max?
We assume $X$ as linearly varied because we have two or three datasets (sample numbers in $X$) for the meteorological variables, and indicators such as interquartile range, STD, or CV cannot be calculated. We do not consider the distributions of measurement samples among these datasets, and used $X$max-$X$min as an alternative to the interquartile range/STD/CV in the calculation of standard measurement uncertainty [1]. Our estimate was in accordance with, and supported by

the concept of measurement uncertainty, where the actual true value is fixed, and the uncertainty range is acquired by the intervals of all measurement samples to represent the precision of a measuring system [2]. As the actual true values of the variables are unknown, we assume the mean of the datasets to be the alternative true values. The actual true values are assumed to lie within the range of the maximum and minimum values of the datasets ($X$max and $X$min) with a linear possibility.

Reference:

[1] White GH. Basics of estimating measurement uncertainty. Clin Biochem Rev. 2008 Aug;29 Suppl 1(Suppl 1): S53-60. PMID: 18852859; PMCID: PMC2556585.

[2] Libretexts. (2023, August 27). 1.3: Measurements, uncertainty and significant figures. Physics LibreTexts. https://phys.libretexts.org/Courses/Georgia_State_University/GSU-TM-Physics_I_(2211)/01%3A_Introduction_to_Physics_and_Measurements/1.03%3A_Measurements_Uncertainty_and_Significant_Figures#:~:text=Discrepancy%20is%20the%20difference%20between%20the%20measured%20value,then%20the%20discrepancy%20of%20the%20values%20is%20high.

**Comment:** Except the location and id, the contributing area can be used as the third criterion for paring the gauge in both GSIM and GSHA. Specifically, if the contributing area are not the same, there is a high probability that not the same gauge is used in GSIM and GSHA for comparison.

Thanks for the suggestion. We have already considered the area difference in our matching process, and we have actually mentioned "the GSIM gauge with a minimum distance and watershed area difference ≤5% to a GSHA gauge was considered" in our original manuscript (Line 377-378).

In addition, I don't understand how the time step impact the annual streamflow indices, e.g., p90 that is reported in Figure R2. Should the estimate of the annual streamflow indices have based on 365 or 366 daily streamflow (if the data is available for the whole year)?

In our calculation, it was not required that streamflow observations are available throughout the year (365 or 366 days), and that observations start and end at 31$^{st}$ Dec. The percentiles were based on the available observations, since we gave several fields for data filtering purposes, including number of days with Q=0, valid observation days, and month with nan>10 days. For instance, for a summer monsoon-controlled watershed in Asia, if NaN values are concentrated in DJF, the p90 and AMF indices are not likely to be influenced. Therefore, it is possible that the missing values and inconsistencies of time span cause some discrepancies in GSIM and GSHA indices in Figure R3 (Figure R2 in last version of reply), but we do not require that the estimate of annual streamflow be based on 365/366 days of data.

[Figure]

Figure R3 Validation of GSHA with GSIM streamflow 90 percentile. The red line is the 1:1 line, while the orange dotted line is the fitting line of the scatter points.